# Synaptic enrichment and dynamic regulation of the two opposing dopamine receptors within the same neurons

**Shun Hiramatsu[1], Kokoro Saito[1], Shu Kondo[2], Hidetaka Katow[3], Nobuhiro Yamagata[4], Chun-Fang Wu[5], Hiromu Tanimoto[1]***

[1]Graduate School of Life Sciences, Tohoku University, Sendai, Japan; [2]Department of Biological Science and Technology, Faculty of Advanced Engineering, Tokyo University of Science, Tokyo, Japan; [3]Department of Cell Biology, New York University, New York, United States; [4]Faculty and Graduate School of Engineering Science, Akita University, Akita, Japan; [5], Department of Biology, University of Iowa, Iowa City, United States

## eLife Assessment

This study uses state-of-the-art methods to label endogenous dopamine receptors in a subset of *Drosophila* mushroom body neuronal types. The authors report that Dop1R1 and Dop2R receptors, which have opposing effects on intracellular cAMP, are present in axons termini of Kenyon cells, as well as those of two classes of dopaminergic neurons that innervate the mushroom body indicative of autocrine modulation by dopaminergic neurons. Additional experiments showing opposing effects of starvation on Dop1R1 and Dop2R levels in mushroom body neurons are consistent with a role for dopamine receptor levels increasing the efficiency of learned food-odour associations in starved flies. Supported by **solid** data, this is an **important** contribution to the field.

***For correspondence:**
hiromut@tohoku.ac.jp

**Abstract** Dopamine can play opposing physiological roles depending on the receptor subtype. In the fruit fly *Drosophila melanogaster*, *Dop1R1* and *Dop2R* encode the $D_1$- and $D_2$-like receptors, respectively, and are reported to oppositely regulate intracellular cAMP levels. Here, we profiled the expression and subcellular localization of endogenous Dop1R1 and Dop2R in specific cell types in the mushroom body circuit. For cell-type-specific visualization of endogenous proteins, we employed reconstitution of split-GFP tagged to the receptor proteins. We detected dopamine receptors at both presynaptic and postsynaptic sites in multiple cell types. Quantitative analysis revealed enrichment of both receptors at the presynaptic sites, with Dop2R showing a greater degree of localization than Dop1R1. The presynaptic localization of Dop1R1 and Dop2R in dopamine neurons suggests dual feedback regulation as autoreceptors. Furthermore, we discovered a starvation-dependent, bidirectional modulation of the presynaptic receptor expression in the protocerebral anterior medial (PAM) and posterior lateral 1 (PPL1) clusters, two distinct subsets of dopamine neurons, suggesting their roles in regulating appetitive behaviors. Our results highlight the significance of the co-expression of the two opposing dopamine receptors in the spatial and conditional regulation of dopamine responses in neurons.

## Introduction

Neurotransmitters typically have multiple cognate receptors, and they may recruit different second messenger systems. Therefore, the expression and localization of receptor subtypes are critical for

determining cellular responses to neurotransmitter inputs. The dopaminergic system offers an ideal in vivo study case to this end, as it regulates a wide array of physiological functions through combinations of different receptor subtypes. In mammals, $D_1$-like receptors are coupled with $G\alpha_s$, thereby activating adenylate cyclase upon ligand binding, whereas $G\alpha_i$-coupled $D_2$-like receptors inhibit cyclase activity (*Beaulieu and Gainetdinov, 2011*). Four dopamine receptors have been identified in *Drosophila*: *Dop1R1*, *Dop1R2*, *Dop2R*, and *DopEcR* (*Han et al., 1996*; *Hearn et al., 2002*; *Srivastava et al., 2005*; *Sugamori et al., 1995*). *Dop1R1* and *Dop2R*, also known as *DopR1*, *dDA1*, and *dumb*, and as *DD2R*, respectively, correspond to the $D_1$- and $D_2$-like receptors, respectively (*Hearn et al., 2002*; *Sugamori et al., 1995*). Dop1R2 and DopEcR are invertebrate-specific and have been reported to recruit different second messenger systems (*Han et al., 1996*; *Srivastava et al., 2005*). Intriguingly, recent data of single-cell RNA-seq and transgenic expression profiling revealed that the expression of these dopamine receptors is highly overlapping in the fly brain (*Croset et al., 2018*; *Davie et al., 2018*; *Kondo et al., 2020*), unlike the spatially segregated expression of the $D_1$- and $D_2$-like receptors in vertebrate brains (*Gerfen and Surmeier, 2011*). Considering the opposing physiological roles of the Dop1R1 and Dop2R, their protein localization, especially in those cells where they are co-expressed, should be critical in determining the responses to the dopamine inputs.

*Drosophila* mushroom bodies (MB) have long served as a unique dopaminergic circuit model to study adaptive behaviors, such as associative learning. MB-projecting neurons and their connections have been systematically described at both mesoscopic and ultrastructural resolutions (*Aso et al., 2014a*; *Li et al., 2020*; *Takemura et al., 2017*; *Tanaka et al., 2008*). Kenyon cells (KCs), the major MB intrinsic neurons, encode a variety of sensory information (*Honegger et al., 2011*; *Turner et al., 2008*; *Vogt et al., 2014*). Intriguingly, each KC receives synaptic inputs from different dopaminergic projections in multiple spatially segmented compartments along its axon in the MB lobe (*Aso et al., 2014a*; *Tanaka et al., 2008*). MB-projecting dopamine neurons (DANs) originate from the three cell clusters (*Mao and Davis, 2009*; *Nässel and Elekes, 1992*). DANs in the protocerebral posterior lateral 1 (PPL1) cluster project to the vertical lobes and the peduncle of the MB, and they control different aspects of associative memory (*Aso et al., 2010*; *Aso et al., 2012*; *Aso and Rubin, 2016*; *Claridge-Chang et al., 2009*; *Krashes et al., 2009*; *Mao and Davis, 2009*; *Masek et al., 2015*; *Riemensperger et al., 2005*; *Takemura et al., 2017*; *Vogt et al., 2014*). The protocerebral anterior medial (PAM) cluster, the largest DAN cluster, mostly projects to the medial lobes of the MB, and many PAM neurons are involved in reward processing (*Burke et al., 2012*; *Felsenberg et al., 2018*; *Huetteroth et al., 2015*; *Ichinose et al., 2021*; *Lin et al., 2014*; *Liu et al., 2012*; *Yamagata et al., 2015*; *Yamagata et al., 2016*). DANs in PPL2ab project to the MB calyx and control the conditioned odor response (*Boto et al., 2019*). In addition to the variety of the dopamine sources, KCs express all four dopamine receptor subtypes (*Deng et al., 2019*; *Kondo et al., 2020*). Besides KCs, these lobe-projecting DANs have synaptic outputs to multiple types of neurons, including MBONs, APL, and DPM (*Li et al., 2020*; *Takemura et al., 2017*; *Zhou et al., 2019*). Given the multitude of modulatory effects of dopamine in the MB (*Berry et al., 2018*; *Cohn et al., 2015*; *Handler et al., 2019*), receptor localization in each cell type provides important information for interpreting such functional diversity.

The projections and synapses of the neurons in the MB circuit are tightly intertwined (*Li et al., 2020*; *Takemura et al., 2017*). Therefore, conventional immunohistochemical approaches using light microscopy do not allow identification of cells from which immunoreactive signals originate. Precise determination of their subcellular localization requires conditional visualization of the proteins of interest only in the target MB neurons (*Bonanno et al., 2024*; *Fendl et al., 2020*; *Sanfilippo et al., 2024*). Employing the CRISPR/Cas9-mediated split-GFP tagging strategy (*Kamiyama et al., 2016*; *Kamiyama et al., 2021*; *Kondo et al., 2020*), we profiled the spatial distribution of endogenous Dop1R1 and Dop2R proteins in KCs, the PAM, and the PPL1 DANs.

## Results

### Co-expression of *Dop1R1* and *Dop2R* genes in the adult *Drosophila* brain

To compare the expression of different dopamine receptor genes in detail, we used T2A-GAL4 knock-ins of the endogenous *Dop1R1* and *Dop2R* genes (*Kondo et al., 2020*) with fluorescent reporters. Both lines labeled many neuropils including the MB (*Figure 1A and B*), and the overlapping expression of

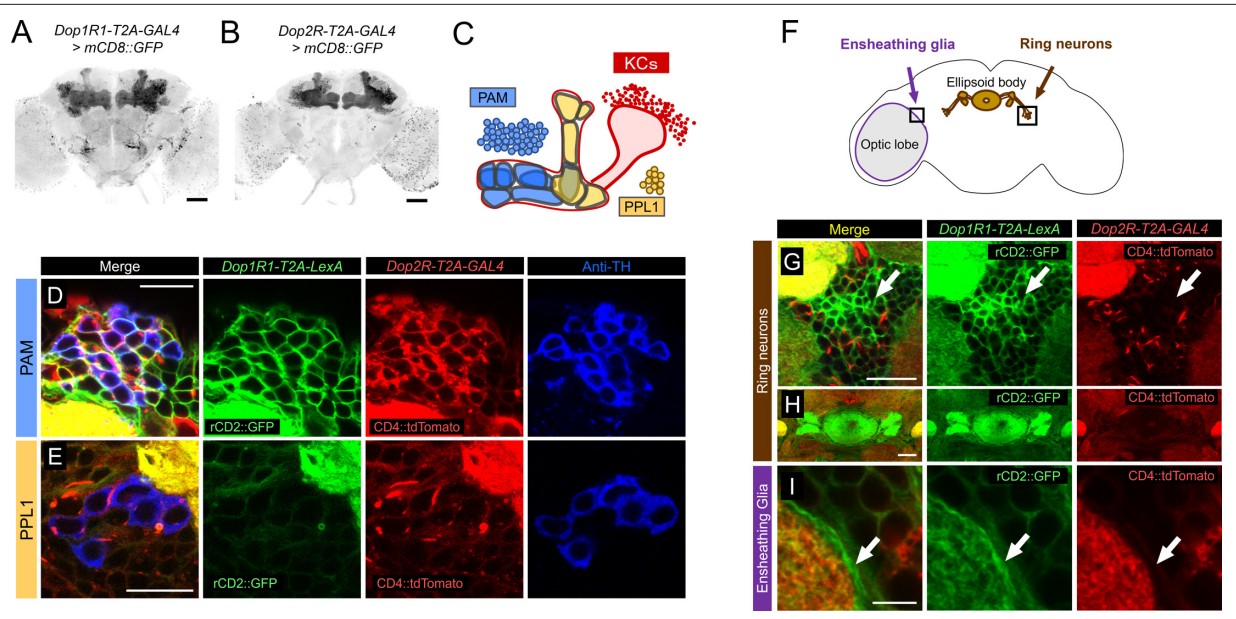

**Figure 1.** Co-expression of *Dop1R1* and *Dop2R* genes in adult *Drosophila* brain. (**A and B**) The expression of *Dop1R1-T2A-GAL4* and *Dop2R-T2A-GAL4* visualized by *UAS-mCD8::GFP*. Maximum-intensity projections of the whole brain. (**C**) Schematic of the Kenyon cells (KCs) and the mushroom bodies (MB)-innervating dopamine neurons from the protocerebral anterior medial (PAM) and posterior lateral 1 (PPL1) clusters. (**D–E, G–I**) Double labeling of *Dop1R1-T2A-LexA* and *Dop2R-T2A-GAL4* expressions by *lexAop-rCD2::GFP* (green) and *UAS-CD4::tdTomato* (red), respectively. Dopamine neurons were immunostained with anti-TH antibody (blue). Single optical sections are shown. Cell bodies of the PAM cluster (**D**), the PPL1 cluster (**E**), ring neurons projecting to the ellipsoid body (**G and H**), and ensheathing glia (**I**) are shown. (**F**) Schematic of the regions shown in (**G–I**). Scale bars, 50 μm (**A and B**), 5 μm (**D, E, and I**), 20 μm (**G and H**).

The online version of this article includes the following figure supplement(s) for figure 1:

**Figure supplement 1.** Co-expression of Dop1R1 and Dop2R genes in Kenyon cells.

these receptors is consistent with our previous quantification of GAL4-positive cells for both genes (58,049 and 68,528 for *Dop1R1* and *Dop2R*, respectively, out of 118,331 brain cells; *Kondo et al., 2020*). Double labeling of *Dop1R1-T2A-LexA* and *Dop2R-T2A-GAL4* expression revealed cells with overlapping and differential patterns (*Figure 1C–I*; see also *Kondo et al., 2020*). We confirmed the co-expression of *Dop1R1* and *Dop2R* in the KCs (*Figure 1—figure supplement 1*) as reported previously (*Kondo et al., 2020*). The PAM cluster of DANs expressed both *Dop1R1* and *Dop2R* (*Figure 1C and D*). On the other hand, most of the DANs in the PPL1 cluster strongly expressed *Dop2R*, but *Dop1R1* only weakly (*Figure 1C and E*). We further found that *Dop1R1*, but not *Dop2R*, was highly expressed in the ring neurons projecting to the ellipsoid body (*Figure 1G and H*; *Hanesch et al., 1989*) and the neuropil ensheathing glia (*Figure 1I*; *Awasaki et al., 2008*). In conclusion, *Dop1R1* and *Dop2R* genes are co-expressed in PAM neurons and KCs but have selective expressions in other cell types.

## Quantification of subcellular enrichment of endogenous proteins in target cells

The overlapping expression of *Dop1R1* and *Dop2R* genes prompted us to examine the subcellular localization of these receptor proteins. To elucidate the localization of these broadly expressed receptors (*Figure 1*), we utilized split-GFP tagging of endogenous proteins (*Kamiyama et al., 2016*; *Kondo et al., 2020*). By adding seven copies of GFP$_{11}$ tags to the C-termini of the Dop1R1 and Dop2R proteins, their intracellular distribution can be visualized specifically in cells expressing GFP$_{1-10}$ through split-GFP reconstitution (*Figure 2A*). To verify the functional integrity of reconstituted GFP (rGFP)-fused receptors, we examined aversive olfactory memory of homozygous flies carrying GFP$_{11}$-tagged dopamine receptors and induced GFP reconstitution in KCs. Both Dop1R1 and Dop2R have been shown to be required for aversive memory in α/β and γ KCs (*Kim et al., 2007*; *Scholz-Kornehl and*

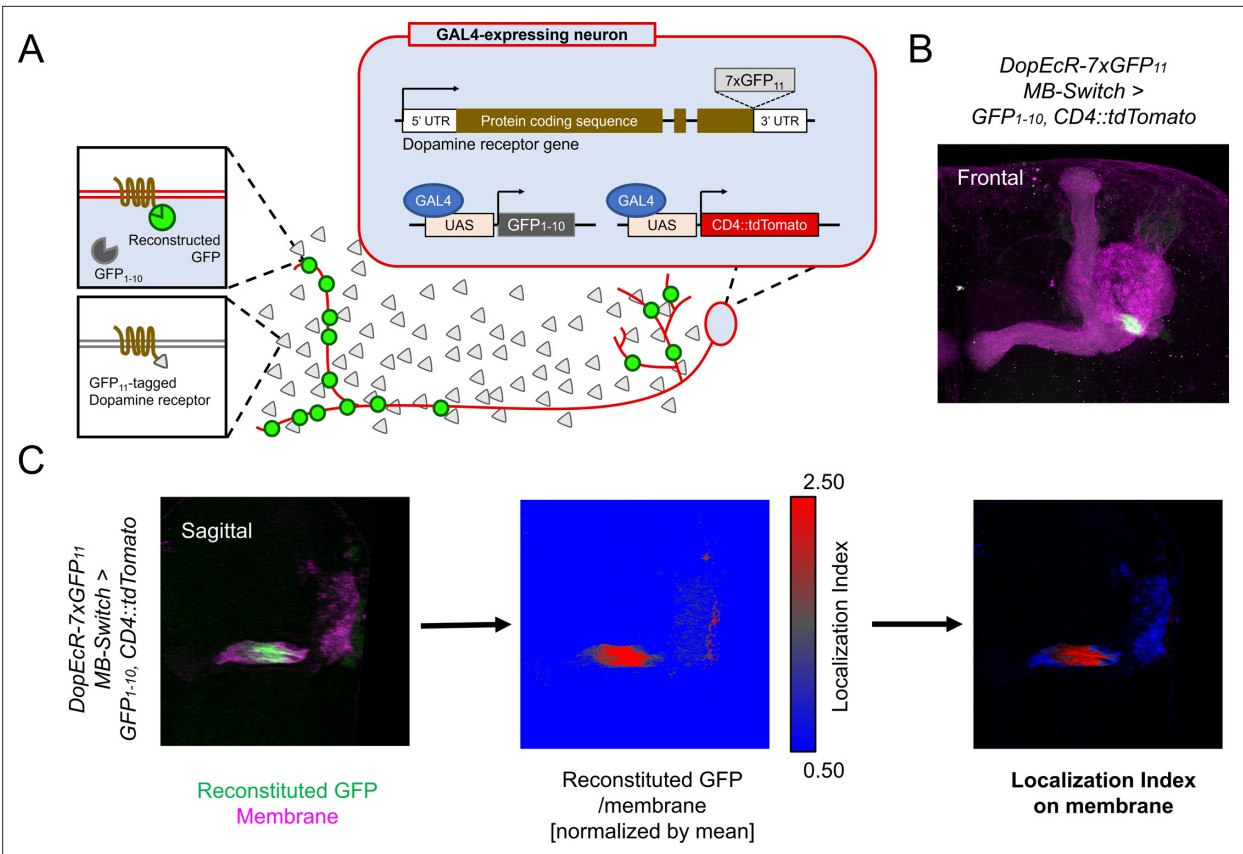

**Figure 2.** Cell-type-specific visualization of endogenous proteins with GFP$_{11}$ tag. (**A**) Principle of cell-type-specific fluorescent labeling of target proteins by GFP$_{11}$ tag. Seven copies of GFP$_{11}$ are fused to the C-terminal of endogenous receptors. GFP$_{1-10}$ and membrane marker CD4::tdTomato are expressed in the target cells by GAL4/UAS system. In the target cells, reconstitution of GFP occurs on the endogenous proteins tagged with GFP$_{11}$. (**B**) As an example, DopEcR::GFP$_{11}$ is visualized in Kenyon cells (KCs) using *MB-Switch*, a ligand-inducible GAL4 driver. To activate Gene-Switch, flies were fed with food containing 200 μM RU486 for 12 hr before dissection. A merged image of reconstituted GFP (green) and cellular membrane visualized by CD4::tdTomato (magenta). Maximum-intensity projection of the whole left mushroom bodies (MB). (**C**) The workflow for visualizing subcellular protein enrichment by localization index (LI). A single sagittal section of the MB calyx and peduncle is shown. The ratio of reconstituted GFP to membrane signal is calculated and normalized by the mean of all voxels to provide LI. In the middle image, LI is color-coded so that red represents local receptor enrichment. In the right image, the intensity of LI color is adjusted based on the membrane signal.

The online version of this article includes the following figure supplement(s) for figure 2:

**Figure supplement 1.** GFP$_{11}$ tagging on Dop1R1 and Dop2R do not affect aversive olfactory memory.

*Schwärzel, 2016*). Memory scores of the rGFP-fused receptor flies were comparable to that of control flies without GFP$_{11}$ insertion (*Figure 2—figure supplement 1*).

To quantify the subcellular enrichment of the receptors, we devised localization index (LI). Briefly, LI is the normalized ratio of the rGFP signal to the reference membrane marker signal (CD4::tdTomato). If the target and reference proteins had the identical distribution, LI would be 1 everywhere. More or less enrichment of rGFP would result in the LI larger or smaller than 1, respectively (*Figure 2C*; see also Materials and methods for details). For visualization, the reference membrane signal was color-coded with LI (*Figure 2C*). As proof of principle, mapping the LI of DopEcR::rGFP signals in KCs highlighted enrichment in the proximal peduncle (*Figure 2B and C*), which is consistent with the previous report (*Kondo et al., 2020*). This representation thus visualizes the subcellular enrichment of the targeted receptors in the plasma membrane of GAL4-expressing cells.

## Colocalization of Dop1R1 and Dop2R proteins

First, we compared the localization of Dop1R1 and Dop2R proteins in KCs, where both receptor genes were highly expressed (*Figure 1—figure supplement 1*). These receptors were distributed throughout KC membranes with predominant enrichment in the lobes (*Figure 3A–C*), but sparsely in the calyx

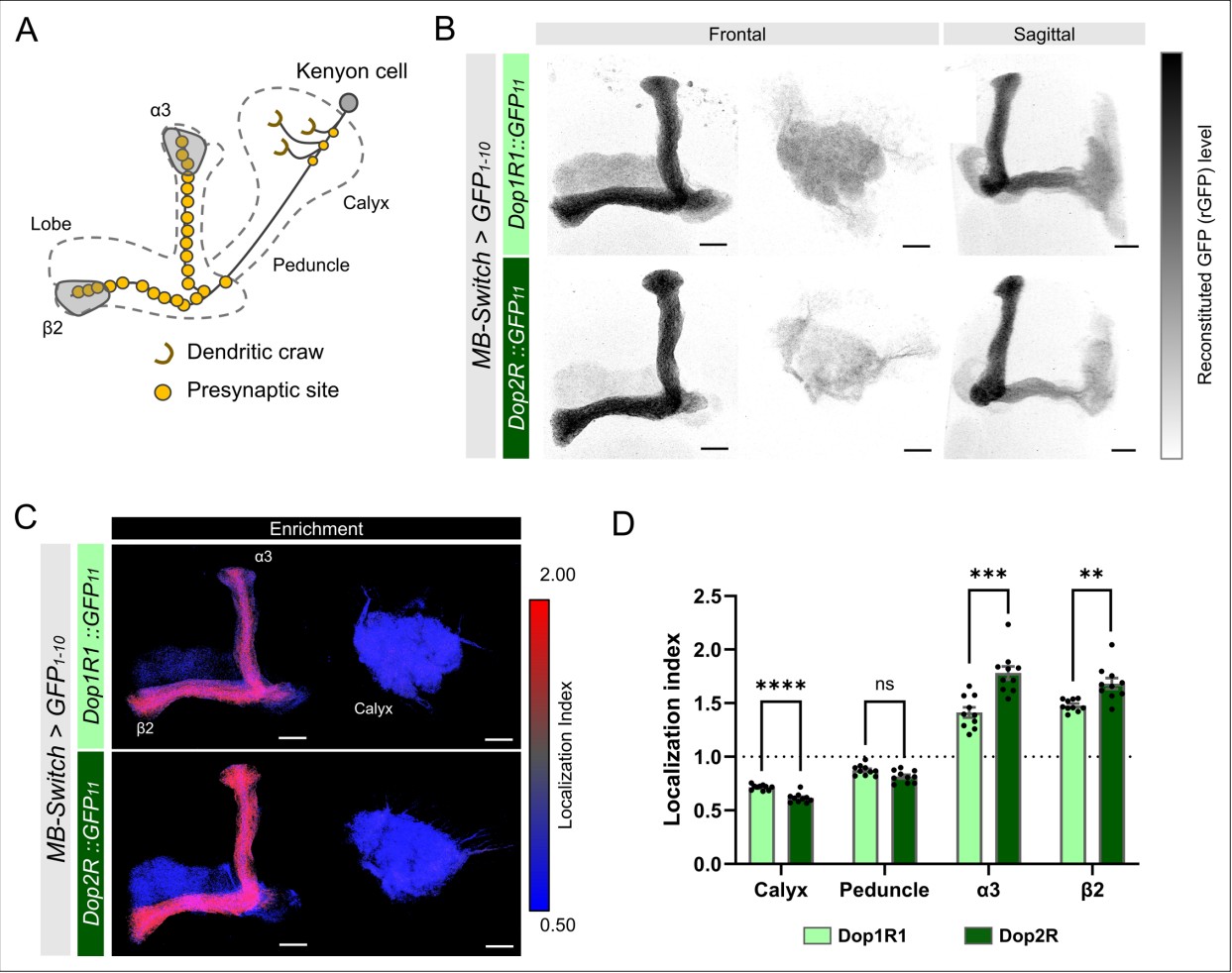

**Figure 3.** Subcellular localization of Dop1R1 and Dop2R in the Kenyon cells (KCs). Subcellular localization of Dop1R1 and Dop2R in the KCs is visualized by GFP_{11} tag. *MB-Switch* was used to express GFP_{1-10} and CD4::tdTomato in the KCs. To activate Gene-Switch, flies were fed with food containing 200 µM RU486 for 72 hr before dissection. (**A**) Schematic showing the projection pattern of an α/β KC. (**B**) Enrichment of Dop1R1 and Dop2R in the mushroom bodies (MB) lobe. Maximum-intensity projections of the lobe (left) and the calyx (middle) are shown in frontal view. The whole left MB are shown in sagittal view (right). Reconstituted GFP signals for both Dop1R1:: and Dop2R::GFP_{11} distributed throughout the MB lobe and the calyx. (**C**) Visualization by localization index (LI) showed more pronounced enrichment of Dop2R than Dop1R1 in the lobe. (**D**) Mean LI of Dop1R1 and Dop2R in the calyx, the peduncle, the α3 and β2 compartment in the lobe. Student's t-test was performed to compare LI of Dop1R1 and Dop2R in each region (N = 10). Error bars; SEM. p>0.05, **p<0.01, ***p<0.001, ****p<0.0001, ns: not significant p>0.05. Scale bars, 20 µm (**B and C**).

The online version of this article includes the following figure supplement(s) for figure 3:

**Figure supplement 1.** Localization index (LI) remains stable across different experimental batches.

(*Figure 3B and C*). LI quantification revealed that enrichment in the lobes was more pronounced in Dop2R compared to Dop1R1, whereas localization to the calyx was sparser in Dop2R than Dop1R1 (*Figure 3D*). We confirmed that the differential localization of these receptors was consistent across multiple experimental batches conducted on different days (*Figure 3—figure supplement 1*).

KCs have major presynaptic sites in the MB lobes, where Dop1R1 and Dop2R are enriched (*Figure 3A–D*). Therefore, we examined receptor localization with the reference of Brp immunostaining, which labels the active zones (AZ) (*Wagh et al., 2006*). To distinguish the AZ in KCs from those in the other neurons, we co-labeled the plasma membrane of KCs and conducted high-magnification imaging using Airyscan. At this resolution, Brp puncta in KCs and those from non-KCs could be distinguished based on their overlap with the KC membrane marker (*Figure 4B and C*). Interestingly, KC-specific visualization of Dop1R1 and Dop2R proteins revealed signals around the Brp puncta of the same cells, suggesting presynaptic localization (*Figure 4B and C*). Additionally, we found the receptor condensates adjacent to the Brp clusters of non-KCs, suggesting their localization at the postsynaptic

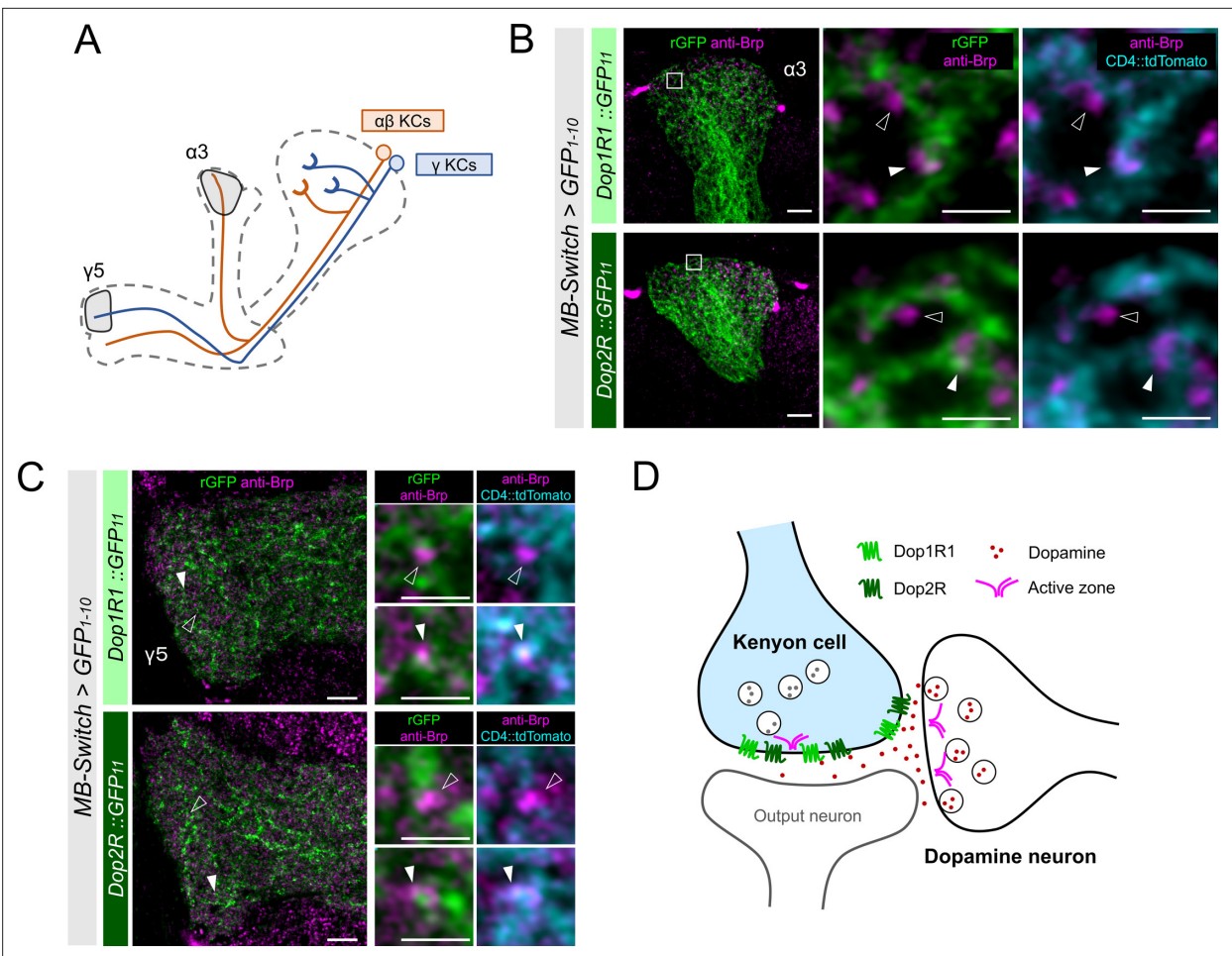

**Figure 4.** High-resolution imaging revealed the two opposing dopamine receptors existing on the presynaptic and postsynaptic sites of Kenyon cells (KCs). (**A**) Schematic showing the projection pattern of α/β and γ KCs. (**B and C**) Airyscan images of Dop1R1::rGFP and Dop2R::rGFP in KCs (green) co-labeled with the active zones (AZ) stained with anti-Brp (magenta). *MB-Switch* was used with 72 hr of RU486 feeding to express GFP$_{1-10}$ and CD4::tdTomato in the KCs. Brp puncta that overlap with CD4::tdTomato signals (cyan) are identified to be presynaptic sites of in KCs (white arrowheads), and those do not overlap are determined to be presynaptic sites of non-KCs (outlined arrows). The synaptic localization of these receptors is similar in the α3 (**B**) and γ5 (**C**) compartments. In the right panels, white squared regions in the left panels are magnified. Scale bars, 5 μm (left), 1 μm (right). (**D**) Illustration of localization of Dop1R1 and Dop2R to presynaptic and postsynaptic sites in the axon terminal of KCs.

sites (*Figure 4B and C*). The synaptic receptor localization is consistent in different subsets of KCs (α/β and γ; *Figure 4A–C*). In conclusion, both of these antagonizing dopamine receptors are enriched in the presynaptic and postsynaptic sites of KCs (*Figure 4D*).

To further clarify the presynaptic localization in KCs, we labeled the AZ of KCs by expressing *Brp$^{short}$::mStraw* (*Fouquet et al., 2009*) and confirmed that both Dop1R1::rGFP and Dop2R::rGFP were associated with the Brp puncta in the lobes (*Figure 5A*, *Figure 5—figure supplement 1*). Interestingly, we found that not all Brp puncta of KCs were associated with the dopamine receptors (*Figure 5A*), suggesting that dopaminergic presynaptic modulation is heterogeneous across release sites. This heterogeneity can well explain differential learning-induced plasticity across boutons within single KCs (*Bilz et al., 2020*; *Davidson et al., 2023*).

To better resolve the presynaptic localization of Dop1R1 and Dop2R, we turned to 'giant' *Drosophila* neurons differentiated from cytokinesis-arrested neuroblasts in culture (*Wu et al., 1990*). The expanded size of the giant neurons is advantageous for investigating the microanatomy of neurons in isolation (*Saito and Wu, 1991*; *Wu et al., 1990*; *Yao et al., 2000*). Importantly, these giant neurons exhibit characteristics of mature neurons, including firing patterns (*Wu et al., 1990*; *Yao and Wu, 2001*; *Zhao and Wu, 1997*) and acetylcholine release (*Yao et al., 2000*), both of which are regulated by cAMP and CaMKII signaling (*Yao et al., 2000*; *Yao and Wu, 2001*; *Zhao and Wu, 1997*). In the

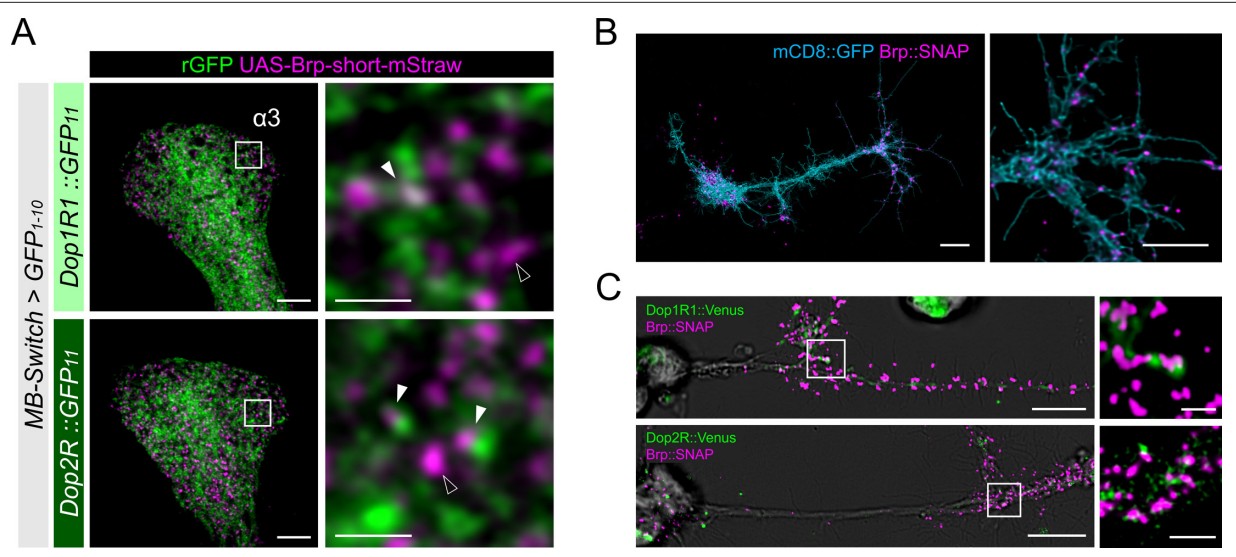

**Figure 5.** Presynaptic localization of Dop1R1 and Dop2R in Kenyon cells (KCs) and giant neurons. (**A**) Double labeling of dopamine receptors (green) and the active zones (AZ) of the KCs (magenta). *MB-Switch* was used with 72 hr of RU486 feeding to express GFP$_{1-10}$ and Brp$^{short}$::mStraw in the KCs. Single focal slices at the α3 compartment are shown. White squares in the left panels are magnified in the right panels. The Brp puncta in KCs were either abutted by the dopamine receptor signals (white arrowheads) or had barely detectable signals nearby (outlined arrowheads). Scale bars, 5 μm (left), 1 μm (right). (**B**) Punctate Brp expression in a giant neuron culture differentiated from cytokinesis-arrested neuroblasts of *OK371-GAL4/UAS-mCD8::GFP* embryos. Aggregated Brp condensates (magenta) were observed in the neurite terminals of the cells marked with mCD8::GFP (cyan) in the right panel. Scale bars, 20 μm (left), 10 μm (right). (**C**) Double labeling of dopamine receptors (green) and the AZs (magenta). *Dop1R1::Venus* or *Dop2R::Venus* was crossed with *Brp::SNAP*. In the left panels, giant neurons extending their neurites from the cell body on the left to the right. In the right panels, white squared regions in the left panels are magnified. Scale bars, 10 μm (left), 2 μm (right).

The online version of this article includes the following figure supplement(s) for figure 5:

**Figure supplement 1.** Presynaptic localization of Dop1R1 and Dop2R in γ Kenyon cells (KCs).

giant neurons from the *Brp::SNAP* embryos (*Kohl et al., 2014*), Brp was localized to the terminals of neurites, but rarely in the proximal neurites (*Figure 5B*). Furthermore, we found punctate Brp clusters in the giant neuron terminals (*Figure 5B*), together recapitulating the essential characteristics of the AZ cytomatrix in adult neurons. To confirm if dopamine receptors localize to these presynaptic sites, we generated the giant neurons from embryos carrying the Venus insertion to *Dop1R1* and *Dop2R* (*Kondo et al., 2020*) together with *Brp::SNAP*. Both Dop1R1 and Dop2R were expressed in the giant neurons and enriched in the same distal axonal segments (*Figure 5C*). A closer investigation revealed that these receptors are associated with the Brp clusters (*Figure 5C*). These observations in the giant neurons are strikingly similar to those in the KCs (*Figures 3, 4, and 5A*), corroborating that the presynaptic localization of these receptors is independent of the circuit context.

## Distinct subcellular enrichment of dopamine receptors in MBONs and DANs

Presynaptic and postsynaptic localization of Dop1R1 and Dop2R in KCs prompted us to investigate MBONs, that have profound postsynaptic contacts with DANs in the MB compartments (*Li et al., 2020*; *Takemura et al., 2017*). MBON-γ1pedc>αβ (also known as MB-MVP2, MBON-11; *Aso et al., 2014a*; *Tanaka et al., 2008*) has most of its postsynaptic sites on the γ1 compartment and the peduncle of the α/β neurons and send axonal projections to the α and β lobes (*Figure 6*; *Dorkenwald et al., 2023*; *Schlegel et al., 2024*). We analyzed the subcellular localization of Dop1R1::rGFP and Dop2R::rGFP in MBON-γ1pedc>αβ by driving GFP$_{1-10}$ using *R83A12-GAL4* (*Perisse et al., 2016*). Strikingly, both Dop1R1 and Dop2R are enriched in the dendritic projection in the γ1 compartment (*Figure 6B and C*) in sharp contrast to the sparse distribution of these receptors in the dendrites of KCs (*Figure 3B–D*). This localization is consistent with previous reports of functional DAN>MBON synapses (*Takemura et al., 2017*) and dopaminergic plasticity on the dendrites (*Boto et al., 2019*; *Pribbenow et al.,*

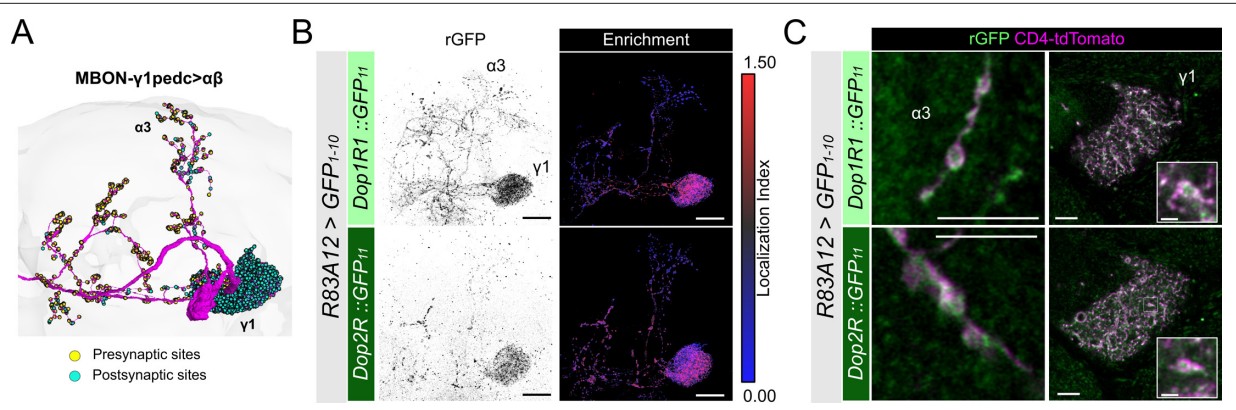

**Figure 6.** Subcellular localization of Dop1R1 and Dop2R in MBON-γ1pedc>αβ. (**A**) The projection pattern of MBON-γ1pedc>αβ from the tracing data in FlyWire (*Dorkenwald et al., 2023*; *Schlegel et al., 2024*). (**B and C**) *R83A12-GAL4* was used to express *UAS-GFP₁₋₁₀* and *UAS-CD4::tdTomato* in the protocerebral anterior medial (PAM) neurons. (**B**) Reconstituted GFP signals (left) of *Dop1R1::GFP₁₁* and *Dop2R::GFP₁₁* in MBON-γ1pedc>αβ. Maximum-intensity projections of the left mushroom bodies (MB) lobe. Visualization of localization index (LI) (right) revealing that both Dop1R1 and Dop2R are enriched in the dendritic projection of MBON-γ1pedc>αβ in the γ1 compartment as well as in the presynaptic boutons. (**C**) Airyscan images of the presynaptic boutons around α3 (left) and dendritic projections in the γ1 compartment (right). White squares in the right panels are magnified in the insertion to show the swelling membrane structures with punctate localization of dopamine receptors. Scale bars, 20 μm (**B**), 5 μm (**C**), 1 μm (**C**, insertion).

*2022*). Additionally, we detected these receptors in the presynaptic boutons (*Figure 6C*), which are consistent with the results in KCs and in the giant neurons (*Figures 4B, C, 5A and C*).

D₂-like receptors in mammals are expressed in DANs and act as autoreceptors, which mediates feedback signals by receiving neurotransmitters released from the neuron on which the receptors reside (*Ford, 2014*). In *Drosophila*, multiple dopamine receptor genes are expressed in DANs (*Figure 1D and E*; *Aso et al., 2019*; *Deng et al., 2019*; *Kondo et al., 2020*), but it is unclear if these receptors function as autoreceptors. We therefore examined the subcellular localization of Dop1R1 and Dop2R in the PAM cluster DANs with a particular focus on their presynaptic terminals. The PAM neurons are polarized; presynaptic proteins are abundantly enriched in the MB lobe and barely detected in dendrites (*Figure 7A and B*). We visualized Dop1R1 and Dop2R proteins in the PAM cluster DANs using *R58E02-GAL4*, and both were localized to the terminals in the MB lobes (*Figure 7C*). Representation of LI for Dop1R1 and Dop2R in the PAM neurons again showed stronger presynaptic enrichment of Dop2R than Dop1R1 (*Figure 7C*). Dop2R was strongly enriched at β'1 compartment showing significantly higher LI than Dop1R1 (*Figure 7D and E*). Higher magnification revealed the accumulation of both Dop1R1 and Dop2R in the boutons (*Figure 7F*). The presynaptic localization of Dop1R1 and Dop2R in DANs suggests that both receptors mediate feedback regulation.

## State-dependent and bidirectional modulation of dopamine receptor expression in the PAM and PPL1 DANs

The activity of MB-projecting DANs is reported to be dynamic and sensitive to feeding states (*Ichinose et al., 2017*; *Liu et al., 2012*; *Plaçais and Preat, 2013*; *Senapati et al., 2019*; *Siju et al., 2020*; *Tsao et al., 2018*; *Yamagata et al., 2016*). We therefore examined if starvation alters the protein expression of Dop1R1 and Dop2R in these MB-projecting DANs (*Figure 8A*). We found a significant elevation of Dop1R1 in the terminals of PAM-γ5 upon starvation for 10 hr or longer (*Figure 8B and C*). In contrast, starvation did not increase Dop2R in PAM-γ5 but rather tended to decrease, if at all (*Figure 8B and C*). We found similar starvation-dependent changes in Dop1R1 and Dop2R levels in other PAM neurons (*Figure 8—figure supplement 1A–D*). These results together suggest that starvation enhances presynaptic dopamine signaling in the reward-related PAM neurons by shifting the balance of Dop1R1 and Dop2R.

DANs of the PAM and PPL1 clusters exert distinct, largely opposite behavioral functions (*Claridge-Chang et al., 2009*; *Liu et al., 2012*). Therefore, plasticity in the PPL1 neurons may differ from that in the PAM neurons. To test this hypothesis, we examined the starvation-dependent change in Dop1R1 and Dop2R protein expression in the PPL1 neurons. To this end, we visualized GFP₁₁-tagged receptors

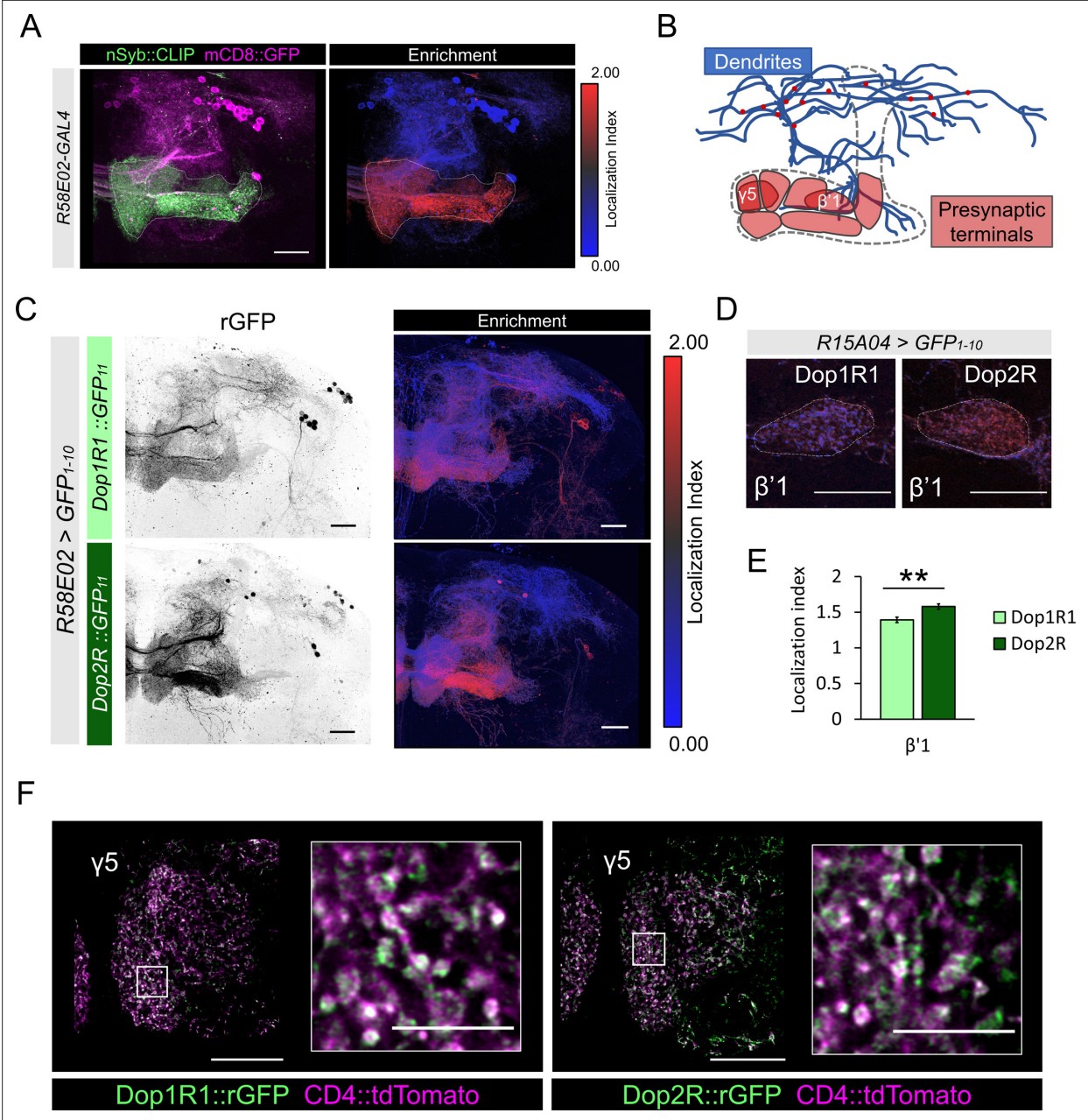

**Figure 7.** Subcellular localization of Dop1R1 and Dop2R in dopamine neurons. (**A**) Maximum-intensity projection image showing the distribution of presynaptic sites in the protocerebral anterior medial (PAM) neurons. Left panel: *R58E02-GAL4* was used to express mCD8::GFP (magenta) and nSyb::CLIP (magenta). Right panel: Visualization by localization index (LI) showing enrichment of nSyb signals in the lobe projection of the PAM neurons. (**B**) Illustrated projection pattern of the PAM neurons. Red puncta on the dendrites indicate the sparse distribution of presynaptic sites in dendrites. (**C–F**) Subcellular localization of GFP$_{11}$-tagged Dop1R1 and Dop2R in the PAM neurons. *R58E02-GAL4* (**C and F**) or *R15A04-GAL4* (**D and E**) was used to express *UAS-GFP$_{1-10}$* and *UAS-CD4::tdTomato* in the PAM neurons. (**C**) Reconstituted GFP signals of *Dop1R1::GFP$_{11}$* and *Dop2R::GFP$_{11}$* in PAM neurons (left). LI visualization revealed the stronger presynaptic enrichment of Dop2R than that of Dop1R1 (right). Maximum-intensity projections of the left hemisphere including the whole mushroom bodies (MB) lobe and dendritic projections of the PAM neurons around the MB. (**D and E**) LI in PAM-β'1 neuron. (**D**) The presynaptic terminals of PAM-β'1 neurons are shown (dashed line). (**E**) Mean LI for Dop1R1 and Dop2R in the β'1 (Mann-Whitney U test, N = 9). Error bars; SEM. (**F**) A single optical slice of the γ5 compartment in the MB lobe obtained using Airyscan. Merged image of reconstituted GFP (green) and CD4::tdTomato (magenta). Insertions are the magnified images of the presynaptic boutons of PAM-γ5 (white squares). Scale bars, 20 μm (**A, C, D, and F**), 5 μm (F, insertion).

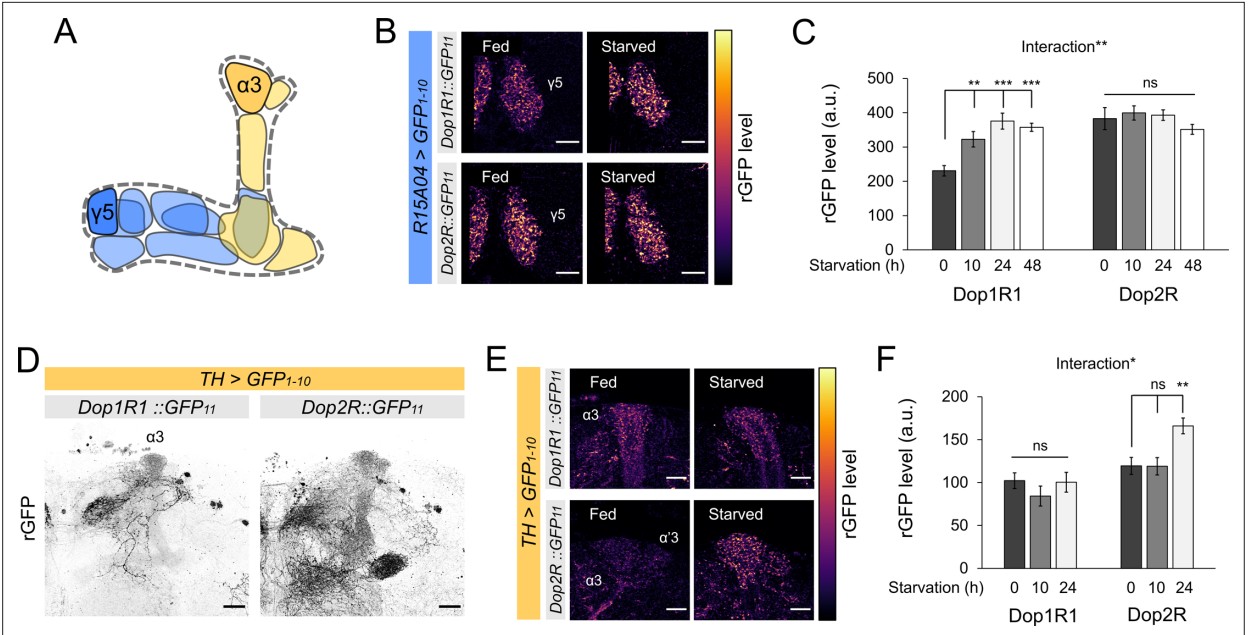

**Figure 8.** Bidirectional modification of dopamine receptor expression in dopamine neurons. (**A**) Schematic illustration of the mushroom bodies (MB) projection of the protocerebral anterior medial (PAM) and posterior lateral 1 (PPL1) dopamine neurons. (**B**) Dop1R and Dop2R in the presynaptic terminals of PAM-γ5 after 48 hr of starvation compared with fed state. (**C**) Quantification of reconstituted GFP (rGFP) signal levels in the presynaptic terminals of PAM-γ5 after 0, 10, 24, and 48 hr of starvation (n = 6–13). (**D**) Reconstituted GFP signals of *Dop1R1::GFP₁₁* and *Dop2R::GFP₁₁* in the PPL1 neurons. In the MB projections of the PPL1 neurons, Dop1R1 was detected in only the α3 compartment. Dop2R was found in all MB projections. Maximum-intensity projections of the MB lobe. (**E**) Dop1R and Dop2R in the presynaptic terminals of PPL1-α3 after 24 hr of starvation compared with fed state. (**F**) Quantification of rGFP signal levels in the presynaptic terminals of PPL1-α3 after 0, 10, and 24 hr of starvation (n = 7–10). Scale bar, 10 μm (**B and E**), 20 μm (**D**). Interaction effects between genotypes and starvation time on protein levels were tested by two-way ANOVA (**C and F**). Bars and error bars represent mean and SEM, respectively (**C and F**). **p<0.01, ***p<0.001, ns: not significant p>0.05.

The online version of this article includes the following figure supplement(s) for figure 8:

**Figure supplement 1.** Starvation-dependent change of dopamine receptors in protocerebral anterior medial (PAM) and posterior lateral 1 (PPL1).

by expressing GFP₁₋₁₀ using *TH-GAL4* (**Friggi-Grelin et al., 2003**). We detected Dop2R proteins in all MB projections of the PPL1 neurons, whereas Dop1R1 proteins were only detectable in the terminals of the PPL1-α3 neuron (**Figure 8D**). Strikingly, the starvation-induced changes in the PPL1 neurons were opposite to those in the PAM: Dop2R, but not Dop1R1, was significantly increased in the α3 compartment (**Figure 8E and F**). In the other compartments, the Dop2R::rGFP levels tended to be higher in starved flies although the increase was not statistically significant (**Figure 8—figure supplement 1E and F**). These results are in line with the state-dependent changes in the physiology of these DANs (**Siju et al., 2020**; **Tsao et al., 2018**). Taken together, starvation induces bidirectional modulation of the dual autoreceptor system in the PPL1 and PAM DANs, and we propose that these changes shift the balance of dopamine output from these subsets (**Figure 9A**).

The dopaminergic signals from the PPL1 neurons inhibit expression of appetitive olfactory memory in fed flies (**Krashes et al., 2009**; **Pavlowsky et al., 2018**). The increased Dop2R autoreceptors in starved flies (**Figure 8E and F**) may thus disinhibit appetitive memory expression by suppressing PPL1 outputs (**Figure 9B**). To test this hypothesis, we examined appetitive memory by transgenic knockdown of Dop2R specifically in the PPL1 cluster neurons using *MB504B-GAL4* (**Vogt et al., 2014**). Indeed, appetitive memory was impaired (**Figure 9C**), suggesting the significance of the negative feedback regulation of the PPL1 via Dop2R in appetitive behavior expression. This result is consistent with our model, which suggests that enhanced negative feedback to PPL1 via Dop2R in the starved state contributes to the expression of appetitive behavior (**Figure 9A and B**).

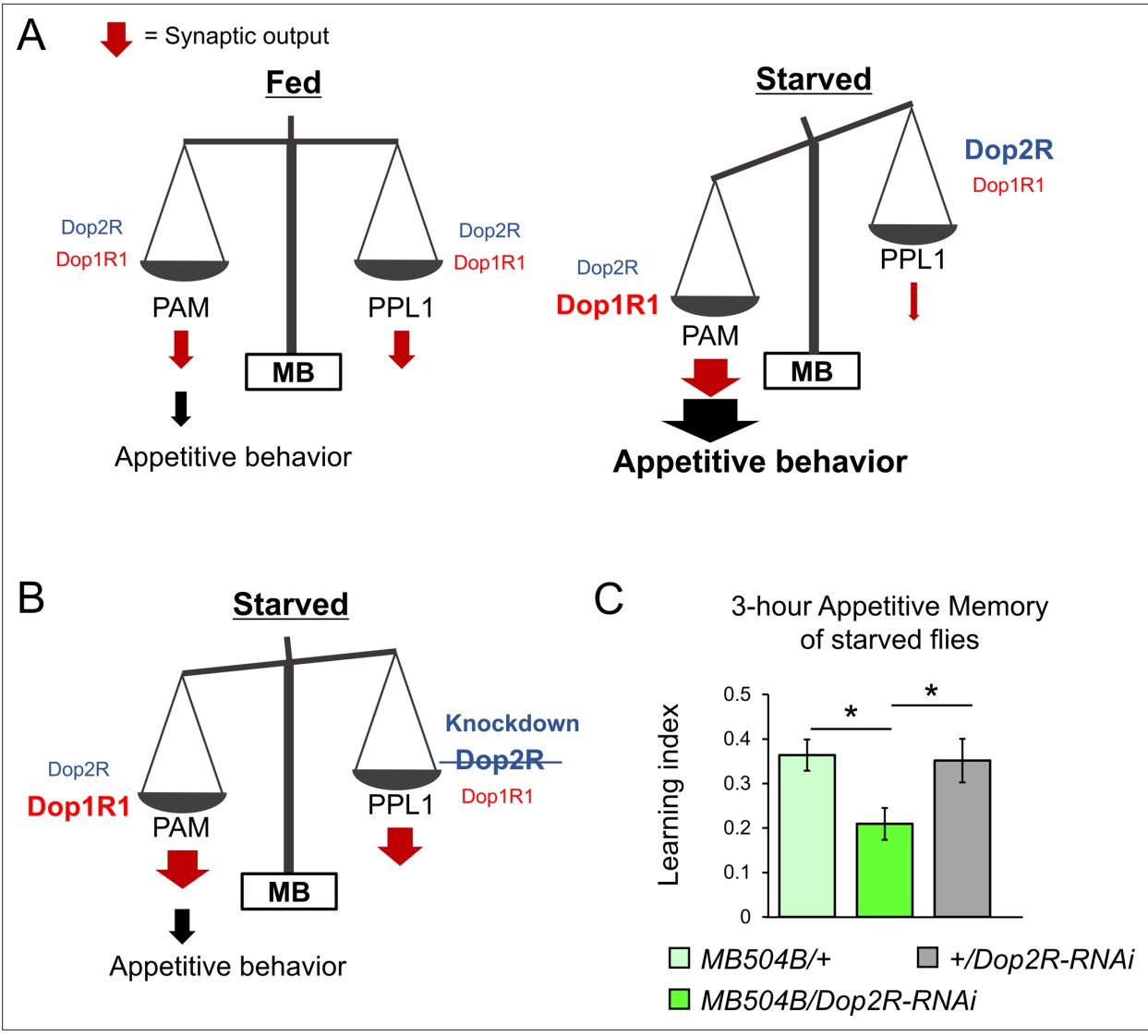

**Figure 9.** The dual dopaminergic feedback regulating starved state-dependent expression of appetitive behavior. (**A**) A working model showing the role of the dual dopaminergic feedback regulation. In the starved state, increased Dop1R1 in protocerebral anterior medial (PAM) neurons and increased Dop2R in posterior lateral 1 (PPL1) neurons changes the balance between the synaptic outputs from these dopamine neurons (DANs) to favor appetitive behavior. (**B**) According to the model, loss of Dop2R in PPL1 upregulates output from PPL1 to attenuate appetitive behavior in starved flies. (**C**) Knockdown of Dop2R in the PPL1 neurons by *MB504B-GAL4* reduced 3 hr appetitive memory performance (t-test with Bonferroni correction, n = 14–15). Bars and error bars represent mean and SEM, respectively. *p<0.05.

## Discussion

The present study demonstrated the expression, subcellular localization, and dynamics of endogenous dopamine receptors, specifically Dop1R1 and Dop2R, in the MB circuit of the fly brain. Compared to intronic insertions of reporters using transposons (*Fendl et al., 2020*), CRISPR/Cas9-based insertion is advantageous in terms of flexibility in fusion sites (*Bonanno et al., 2024*; *Kamiyama et al., 2021*; *Sanfilippo et al., 2024*; *Williams et al., 2019*). This enables the C-terminal tagging of dopamine receptors that do not perturb protein localization and function (*Figure 2—figure supplement 1*).

The split-GFP reconstitution strategy enables sensitive detection of low-abundant endogenous proteins by the tandem multimerization of the GFP$_{11}$ tags, as shown in other epitope tags such as SunTag and smFPs (*Tanenbaum et al., 2014*; *Viswanathan et al., 2015*). The native fluorescence of rGFP was sufficient to detect tagged proteins at the high signal-to-noise ratio, without the need for signal amplification using antibodies. The background fluorescence of the split-GFP fragments was

practically negligible (*Figure 8D*; *Kamiyama et al., 2016*; *Kondo et al., 2020*). This approach can thus be applicable to monitor localization and dynamics of endogenous proteins of low abundance.

Our cell-type-specific receptor labeling offers further potential for ultrastructural examination using higher resolution imaging techniques. For example, synaptic condensates of Dop1R1 and Dop2R at the resolution in this study (*Figures 4 and 5*) did not allow us to distinguish the receptors on the plasma membrane and those that are internalized (*Dumartin et al., 1998*; *Kotowski et al., 2011*). Thus, application of super-resolution fluorescent imaging and expansion microscopy (*Gao et al., 2019*) would disentangle the precise membrane localization of the receptors.

## Spatial regulation of dopamine signaling through the two opposing receptors

How do Dop1R1 and Dop2R function in presynaptic terminals? Released dopamine is not confined to the synaptic cleft, but diffused in the extracellular space (*Liu et al., 2021*; *Rice and Cragg, 2008*). Such volume transmission can increase the number of target synapses by 10 times (*Li et al., 2020*; *Takemura et al., 2017*). Thus, the broader distribution of Dop1R1 (*Figures 3B–D and 7C–E*) may enable it to respond to extrasynaptic dopamine. In response to residual dopamine, Dop2R at the AZ may inhibit activities of adenylate cyclases and voltage-gated calcium channels (*Beaulieu and Gainet-dinov, 2011*; *Hearn et al., 2002*), thereby reducing the noise of the second messengers. Collectively, the spatial configuration of Dop1R1 and Dop2R may enhance the sensitivity and precision of dopaminergic modulation.

Furthermore, presynaptic receptor localization (*Figures 4B–D, 5A, C, 6C, and 7F*) suggests a spatially confined cAMP, forming nanodomains (*Anton et al., 2022*; *Bock et al., 2020*; *Maiellaro et al., 2016*; *Zhang et al., 2020*). cAMP signaling has been shown to regulate multiple events at presynaptic terminals, such as the molecular assembly at the AZ and synaptic vesicle dynamics (*Ehmann et al., 2018*; *Kittel and Heckmann, 2016*; *Kuromi and Kidokoro, 2000*; *Kuromi and Kidokoro, 2005*; *Maiellaro et al., 2016*; *Renger et al., 2000*; *Sachidanandan et al., 2023*). Especially in KCs, cAMP-dependent plasticity underlies associative memory (*Boto et al., 2014*; *Cohn et al., 2015*; *Louis et al., 2018*; *Noyes and Davis, 2023*; *Stahl et al., 2022*). Presynaptic cAMP regulation through Dop1R1 and Dop2R therefore explains the requirement of these receptors in associative memory (*Kim et al., 2007*; *Scholz-Kornehl and Schwärzel, 2016*).

## The dual autoreceptor system may shape dopamine release

Presynaptic localization of Dop1R1 and Dop2R in DANs (*Figure 7C–F*) strongly suggests their functions as autoreceptors. In the *Drosophila* nervous system, Dop2R was shown to negatively regulate dopamine release (*Shin and Venton, 2022*; *Vickrey and Venton, 2011*). Tight presynaptic enrichment in PAM (*Figure 7F*) suggests that Dop2R receives high levels of dopamine and effectively prevents overactivation. The presence of Dop1R1 in DAN terminals (*Figure 7F*) was unexpected and introduces another layer of presynaptic regulations to dopamine release. Although it must be functionally verified, presynaptic Dop1R1 likely provides positive feedback to the dopamine release as an autoreceptor. This positive feedback would be particularly important for signal amplification when extracellular dopamine concentrations are low. Consistent this hypothesis, presynaptic Dop1R1 was undetectable in most PPL1 DANs (*Figure 8D*), which have been reported to have high spontaneous activities (*Feng et al., 2021*; *Plaçais and Preat, 2013*). Taken together, this dual autoreceptor system likely fine-tunes the amplitude and kinetics of dopamine release. Alternatively, these presynaptic receptors could potentially receive extrasynaptic dopamine released from other DANs. Therefore, the autoreceptor functions need to be experimentally clarified by manipulating the receptor expression in DANs.

Dopamine receptor expression is reported to be associated with prolonged exposure to psychoactive substances, such as caffeine and ethanol (*Andretic et al., 2008*; *Kanno et al., 2021*; *Kondo et al., 2020*; *Petruccelli et al., 2018*). Our study further showed starvation-dependent changes of Dop1R1 and Dop2R in DAN terminals (*Figure 8*). Strikingly, starvation responses of presynaptic Dop1R1 and

Dop2R were differential depending on the DAN cell types (*Figure 8*). These results indicate that starvation bidirectionally changes the dual autoreceptor system, putting more weight on the PAM output over PPL1 to control the expression of appetitive behavior (*Figure 9*). This shifted balance explains the state-dependent changes in the presynaptic activity of these two clusters of DANs (*Siju et al., 2020*; *Tsao et al., 2018*) and bidirectional modulation of the MB output (*Aso et al., 2014b*; *Ichinose et al., 2021*; *Owald et al., 2015*).

# Materials and methods

## Key resources table

| Reagent type (species) or resource | Designation | Source or reference | Identifiers | Additional information |
|---|---|---|---|---|
| Antibody | Mouse monoclonal anti-TH | ImmunoStar Inc, Hudson, WI, USA | #22941; RRID:AB_1267100 | IF(1:100) |
| Antibody | Mouse monoclonal anti-Brp | Developmental Studies Hybridoma Bank, Iowa city, IA, USA | nc82 | IF(1:40) |
| Antibody | Rabbit polyclonal anti-DsRed | Takara Bio USA, Inc, San Jose, CA, USA | #632496 | IF(1:2000) |
| Antibody | Goat polyclonal anti-mouse Alexa Fluor 405 | Invitrogen, Waltham, MA, USA | #A31553; RRID:AB_221604 | IF(1:1000) |
| Antibody | Goat polyclonal anti-mouse Alexa Fluor 633 | Invitrogen, Waltham, MA, USA | #A21052; RRID:AB_2535719 | IF(1:400 for anti-TH, 1:200 for anti-Brp) |
| Antibody | Goat polyclonal anti-rabbit Alexa Fluor 568 | Invitrogen, Waltham, MA, USA | #A11036; RRID:AB_10563566 | IF(1:1000) |
| Chemical compound, drug | SNAP-Cell 647-SiR | Ipswich, MA, USA | S9102S | |
| Chemical compound, drug | CLIP-Surface 547 substrate | New England Biolabs Inc, Ispwich, MA, USA | #S9233S | |
| Chemical compound, drug | RU486 (Mifepristone) | Tokyo Chemical Industry Co., Tokyo, Japan | #M1732 | |
| Genetic reagent (*D. melanogaster*) | Dop1R1-T2A-GAL4 | *Kondo et al., 2020* | N/A | |
| Genetic reagent (*D. melanogaster*) | Dop2R-T2A-GAL4 | *Kondo et al., 2020* | N/A | |
| Genetic reagent (*D. melanogaster*) | Dop1R1-T2A-LexA | *Kondo et al., 2020* | N/A | |
| Genetic reagent (*D. melanogaster*) | Dop1R1::7xGFP11 | *Kondo et al., 2020* | N/A | |
| Genetic reagent (*D. melanogaster*) | Dop2R::7xGFP11 | *Kondo et al., 2020* | N/A | |
| Genetic reagent (*D. melanogaster*) | DopEcR::7xGFP11 | *Kondo et al., 2020* | N/A | |
| Genetic reagent (*D. melanogaster*) | Dop1R1::Venus | *Kondo et al., 2020* | N/A | |
| Genetic reagent (*D. melanogaster*) | Dop2R::Venus | *Kondo et al., 2020* | N/A | |
| Genetic reagent (*D. melanogaster*) | UAS-GFP1-10 | *Kondo et al., 2020* | N/A | |
| Genetic reagent (*D. melanogaster*) | UAS-mCD8::GFP | *Pfeiffer et al., 2010* | BDSC #32194 | |
| Genetic reagent (*D. melanogaster*) | lexAop-rCD2::GFP | *Miyamoto et al., 2012* | N/A | |
| Genetic reagent (*D. melanogaster*) | UAS-CD4::tdTomato | *Han et al., 2011* | BDSC #35841 | |
| Genetic reagent (*D. melanogaster*) | UAS-nSyb::CLIP | *Kohl et al., 2014* | BDSC #58398 | |
| Genetic reagent (*D. melanogaster*) | UAS-Brp-short::mStrawberry | *Fouquet et al., 2009* | N/A | |

*Continued on next page*

*Continued*

| Reagent type (species) or resource | Designation | Source or reference | Identifiers | Additional information |
|---|---|---|---|---|
| Genetic reagent (*D. melanogaster*) | *Brp::SNAP* | *Kohl et al., 2014* | BDSC #58397 | |
| Genetic reagent (*D. melanogaster*) | *MB-Switch* | *Mao et al., 2004* | BDSC #81013 | |
| Genetic reagent (*D. melanogaster*) | *OK371-GAL4* | *Mahr and Aberle, 2006* | BDSC #26160 | |
| Genetic reagent (*D. melanogaster*) | *R83A12-GAL4* | *Jenett et al., 2012* | BDSC #40348 | |
| Genetic reagent (*D. melanogaster*) | *R15A04-GA4* | *Jenett et al., 2012* | BDSC #48671 | |
| Genetic reagent (*D. melanogaster*) | *R58E02-GAL4* | *Jenett et al., 2012* | BDSC #41347 | |
| Genetic reagent (*D. melanogaster*) | *TH-GAL4* | *Friggi-Grelin et al., 2003* | N/A | |
| Genetic reagent (*D. melanogaster*) | *MB504B* | *Pfeiffer et al., 2010* | BDSC #68329 | |
| Genetic reagent (*D. melanogaster*) | *UAS-Dop2R-RNAi* | *Perkins et al., 2015* | BDSC #50621 | |
| Software, algorithm | Fiji | *Schindelin et al., 2012*, http://fiji.sc | RRID:SCR_002285 | |
| Software, algorithm | GraphPad Prism 5 | GraphPad Software | RRID:SCR_002798 | |

## Flies

Flies were raised on standard cornmeal food at 25°C under a 12:12 hr light-dark cycle (zeitgeber time [ZT]0 at 8 AM) for all experiments. The GAL4-UAS system was used to express the transgenes of interest in specific neuron subtypes. Flies carrying GAL4 were crossed to another strain carrying UAS reporters, and F1 progenies were used in the experiments. To visualize GFP$_{11}$-tagged dopamine receptors in the specific cell types, female fly strains carrying *UAS-CD4::tdTomato*, *UAS-GFP$_{1-10}$*, and GAL4 driver were crossed to male fly strains carrying *Dop1R1::GFP$_{11}$* or *Dop2R::GFP$_{11}$*, and F1 progenies were used. To make the giant neuron culture, embryos from *Brp::SNAP* or the F1 progeny of *OK371-GAL4* crossed with *Brp::SNAP, UAS-mCD8::GFP, UAS-nSyb::CLIP* flies were used. For the fly lines used in our manuscript, see Key resources table.

## RU486 feeding

To activate Gene-Switch, flies are fed with food containing RU486 (mifepristone). Food containing 200 µM of RU486 was prepared as described previously (*Mao et al., 2004*). In brief, 200 mg of RU486 was dissolved in 10 ml of 99.5% ethanol to make a stock solution. 4.3 µl of the stock solution was added to 1 ml of molten fly food and mixed thoroughly. The molten food with RU486 was poured into vials or bottles and cooled to make solid food.

## Brain dissection and immunohistochemistry

Flies were sorted under CO$_2$ anesthesia to select males with the specific genotypes and kept in a food vial for recovery at least 24 hr prior to the dissection. Fly brains were dissected 3–7 days after eclosion in ice-cold phosphate-buffered saline (PBS). After dissection, brains were kept in ice-cold PBS with 2% paraformaldehyde (PFA) for up to 30 min. For fixation, brains were incubated in 2% PFA in PBS for 1 hr at room temperature. Fixed brains were washed in PBS with 0.3% Triton X-100 (PBT) for 10 min three times. For rGFP imaging, fixed brains were mounted in SeeDB2 (*Ke et al., 2016*), and native fluorescence was imaged without signal amplification.

For chemical tagging reaction of nSyb::CLIP, brains were incubated in PBT containing CLIP-Surface 547 substrate (1 µM) for 15 min at room temperature. Brains were washed in PBT for 10 min three times.

For immunohistochemistry, fixed brains were blocked in 3% normal goat serum (NGS; Sigma-Aldrich, G9023) for 30 min at room temperature. Brains were incubated in primary and secondary antibodies diluted in PBT with 1% NGS over two nights at 4°C, respectively. After each step, brains

were washed three times in PBT for longer than 20 min at room temperature and mounted in 86% glycerol. The following primary antibodies were used: mouse anti-TH (1:100), mouse anti-Brp (1:40), rabbit anti-DsRed (1:2000). Secondary antibodies: Alexa Fluor 633 goat anti-mouse (1:400 for anti-TH, 1:200 for anti-Brp).

## Fluorescent imaging

For image acquisition, whole-mount brains were scanned with the Olympus FV1200 confocal microscope with the following objective lens; ×20 oil (NA = 0.85, UPLSAPO20XO, Olympus; *Figure 1A, B, G, and H*), ×30 silicone (NA = 1.05, UPLSAPO30XS, Olympus; *Figure 3B and C*), ×40 oil (NA = 1.3, UPLFLN40XO, Olympus; *Figures 2B, C, 6B, 7A, C, D, 8B, E*, *Figure 8—figure supplement 1A, C, and E*), or ×60 oil (NA = 1.42, PLAPON60XO, Olympus; *Figure 1D, E, I*, *Figure 1—figure supplement 1*). Z-stack images were acquired.

For high-resolution imaging in *Figures 4B, C, 5A, 6C, 7F*, *Figure 5—figure supplement 1* , we used Airyscan on Zeiss LSM800 with ×63 oil objective lens (NA = 1.40, Plan-Apochromat).

## Image analysis

All image analyses were conducted on Fiji (*Schindelin et al., 2012*).

To visualize the subcellular localization of dopamine receptors in defined neurons, we devised the LI. Relative receptor density was calculated in each voxel by dividing the receptor (rGFP) signals by the corresponding membrane signals (CD4::tdTomato). To set ROI, any voxels devoid of membrane signal were censored. The local density was normalized by the mean values in the ROI. For the visual representation, the reference membrane signals were colored according to the normalized LI (*Figure 2C*). To quantify mean LI in each subcellular region (*Figures 3H and 7E*), ROI was manually set. Each data point represents a single fly brain sample.

## Embryonic giant neuron culture

Multinucleated giant neurons from neuroblasts were generated as described previously (*Wu et al., 1990*). Briefly, the interior of a gastrula from stage 6/7 embryo was extracted with a glass micropipette and dispersed into a drop of culture medium (~40 μm) on an uncoated coverslip. The culture medium contained 80% *Drosophila* Schneider's insect medium (S0146-100ML, Merck KGaA, Darmstadt, Germany) and 20% fetal bovine serum (F2442-100ML, Merck KGaA, Darmstadt, Germany), with the addition of 50 μg/ml streptomycin, 50 U/ml penicillin (all from Sigma, St. Louis, MO, USA), and cytochalasin B (CCB; 2 μg/ml; Sigma, St. Louis, MO, USA). CCB was removed by washing with CCB-free medium 1 day after plating. All cultures were grown in a humidified chamber.

To label Brp::SNAP, cells were incubated in fluorescent SNAP substrate diluted in the culture medium (1:5000, SNAP-Cell 647-SiR) for half an hour at room temperature. Subsequently, the same cells were labeled by the SNAP substrate to minimize the crosstalk. After the incubation, cells were washed with the culture medium and subjected to confocal scanning with Leica TCS SP8 microscopy equipped with a ×40 oil immersion objective (HC PL APO ×40/1.30 Oil PH3 CS2, Leica). Acquired images were then processed with the LIGHTNING software.

## Behavioral assays

The experimental protocols for appetitive learning experiment (*Figure 9C*) were as described previously (*Ichinose and Tanimoto, 2016*). For appetitive conditioning, a group of approximately 50 flies in a training tube alternately received 3-octanol (3OCT; Merck) and 4-methylcyclohexanol (4MCH; Sigma-Aldrich), for 1 min in a constant air flow with or without reward with an interval of 1 min between the two odor presentations. These odorants were diluted to 1.2% and 2% with paraffin wax (Sigma-Aldrich), respectively. Dried 2 M sucrose (Sigma-Aldrich) on a piece of filter paper was used as the reward. Flies were starved in the presence of wet tissue paper for 24 hr before appetitive conditioning. For testing, flies were given a choice between the odor paired with reward (conditioned stimulus, CS+) and another one unpaired (CS–). Their distribution in the plexiglass T-maze was video-recorded one frame per second for 2 min with the CMOS cameras (GS3-U3-51S5M, FLIR) under infrared LED illumination (*Ichinose and Tanimoto, 2016*). Flies were forced to stay on the floor by applying Fluon (Insect-a-Slip PTFE30, BioQuip Products) on the side and top of T-maze arms.

Learning index (LI) was then calculated by counting the number of flies in each arm using ImageJ macro as described previously (*Ichinose and Tanimoto, 2016*) with the following formula (*Tempel et al., 1983*):

$$Learning\ index = \frac{\#_{CS-} - \#_{CS+}}{\#_{CS+} + \#_{CS-}}$$

where $\#_{CS+}$ and $\#_{CS-}$ imply the number of flies in the CS+ and the CS– arms, respectively. LI was calculated for every second and was averaged for the last 60 s of the 120 s test session. An average of a pair of reciprocally trained groups was used as a single data point.

In aversive associative learning experiment (*Figure 2—figure supplement 1*), flies were conditioned with twelve 1.5 s pulses of 90 V electric shock paired with either 4MCH or 3OCT (CS+) for 1 min. After 1 min of interval, another odor (CS–) was presented without electric shock for 1 min. Flies were tested immediately after training. In the test, flies were allowed to choose between the CS+ and CS– odors in a T-maze. The number of flies in each side of the arm (#CS+ and #CS–) after 2 min was used to calculate LI as described above.

## Statistics

For multiple comparison (*Figure 2—figure supplement 1*, *Figure 8C, F*, *Figure 8—figure supplement 1*, and *Figure 9C*), statistics were performed by Prism 5 (GraphPad). Data were always first tested for normality (Shapiro-Wilk test) and for homoscedasticity (Spearman's test or Bartlett's test). If these assumptions were not violated, parametric tests (one- or two-way ANOVA, followed by Dunnett's test or t-test with Bonferroni correction) were applied. If data did not suffice the assumptions for parametric tests, nonparametric tests (Kruskal-Wallis, followed by Dunn's post hoc pairwise test) were performed. For comparison of two groups, Student's t-test (*Figure 3D*, *Figure 3—figure supplement 1*) or Mann-Whitney U test (*Figure 7E*) were performed.

Bars and error bars represent means and SEM, respectively, in all figures. For all figures, significance corresponds to the following symbols: $*p<0.05$; $**p<0.01$; $***p<0.001$; ns: not significant $p>0.05$.

## Additional information

### Competing interests

Hiromu Tanimoto: Reviewing editor, *eLife*. The other authors declare that no competing interests exist.

### Funding

| Funder | Grant reference number | Author |
|---|---|---|
| Support for Pioneering Research Initiated by the Next Generation | JPMJSP2114 | Shun Hiramatsu |
| Japan Society for the Promotion of Science | 22H04829 | Hiromu Tanimoto |
| Japan Society for the Promotion of Science | 20H00519 | Hiromu Tanimoto |
| Japan Society for the Promotion of Science | 22H05481 | Hiromu Tanimoto |
| Japan Society for the Promotion of Science | 22KK0106 | Hiromu Tanimoto |
| Japan Society for the Promotion of Science | 17H01378 | Hiromu Tanimoto |
| Japan Society for the Promotion of Science | 20H03246 | Shu Kondo |
| Japan Society for the Promotion of Science | 19KK0383 | Nobuhiro Yamagata |

| Funder | Grant reference number | Author |
| --- | --- | --- |
| Japan Society for the Promotion of Science | 17H04765 | Nobuhiro Yamagata |
| Takeda Foundation | | Nobuhiro Yamagata |

The funders had no role in study design, data collection and interpretation, or the decision to submit the work for publication.

## Author contributions

Shun Hiramatsu, Conceptualization, Data curation, Formal analysis, Funding acquisition, Investigation, Writing - original draft, Project administration; Kokoro Saito, Data curation, Formal analysis, Investigation, Writing - review and editing; Shu Kondo, Hidetaka Katow, Resources, Methodology, Writing - review and editing; Nobuhiro Yamagata, Data curation, Funding acquisition, Investigation, Writing - review and editing; Chun-Fang Wu, Supervision, Writing - review and editing; Hiromu Tanimoto, Conceptualization, Supervision, Funding acquisition, Writing - original draft, Project administration

## Author ORCIDs

Shun Hiramatsu ⓘ https://orcid.org/0000-0001-8318-4105
Kokoro Saito ⓘ http://orcid.org/0009-0009-1845-5074
Shu Kondo ⓘ http://orcid.org/0000-0002-4625-8379
Hidetaka Katow ⓘ http://orcid.org/0009-0005-0449-230X
Nobuhiro Yamagata ⓘ http://orcid.org/0000-0003-1993-2038
Chun-Fang Wu ⓘ https://orcid.org/0000-0002-4973-2021
Hiromu Tanimoto ⓘ https://orcid.org/0000-0001-5880-6064

Reviewer #1 (Public review): https://doi.org/10.7554/eLife.98358.3.sa1
Reviewer #2 (Public review): https://doi.org/10.7554/eLife.98358.3.sa2
Author response https://doi.org/10.7554/eLife.98358.3.sa3

---

# Additional files

## Supplementary files
MDAR checklist

## Data availability
All data generated or analyzed during this study are included in the manuscript and supplementary materials. The raw datasets have been deposited in GIN (G-Node Infrastructure) services and can be accessed (https://doi.org/10.12751/g-node.i3xmim).

The following dataset was generated:

| Author(s) | Year | Dataset title | Dataset URL | Database and Identifier |
| --- | --- | --- | --- | --- |
| Hiramatsu S, Saito K, Kondo S, Katow H, Yamagata N, Wu CF, Tanimoto H | 2025 | Synaptic enrichment and dynamic regulation of the two opposing dopamine receptors within the same neurons | https://doi.org/10.12751/g-node.i3xmim | G-Node Infrastructure, 10.12751/g-node.i3xmim |

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
