## [Editor Report · eLife Assessment]

This study uses state-of-the-art methods to label endogenous dopamine receptors in a subset of *Drosophila* mushroom body neuronal types. The authors report that Dop1R1 and Dop2R receptors, which have opposing effects on intracellular cAMP, are present in axons termini of Kenyon cells, as well as those of two classes of dopaminergic neurons that innervate the mushroom body indicative of autocrine modulation by dopaminergic neurons. Additional experiments showing opposing effects of starvation on Dop1R1 and Dop2R levels in mushroom body neurons are consistent with a role for dopamine receptor levels increasing the efficiency of learned food-odour associations in starved flies. Supported by **solid** data, this is an **important** contribution to the field.

---

## [Referee Report · Reviewer #1 (Public review)]

Summary:

This is an important and interesting study that uses the split-GFP approach. Localization of receptors and correlating them to function is important in understanding the circuit basis of behavior.

Strengths:

The split-GFP approach allows visualization of subcellular enrichment of dopamine receptors in the plasma membrane of GAL4-expressing neurons allowing for high level of specificity.

The authors resolve the presynaptic localization of DopR1 and Dop2R, in "giant" *Drosophila* neurons differentiated from cytokinesis-arrested neuroblasts in culture as its not clear in the lobes and calyx.

Starvation induced opposite responses of dopamine receptor expression in the PPL1 and PAM DANs provides key insights into models of appetitive learning.

Starvation induced increase in D2R allows for increased negative feedback that the authors test in D2R knockout flies where appetitive memory is diminished.

This dual autoreceptor system is an attractive model for how amplitude and kinetics of dopamine release can be fine tunes and controlled depending on the cellular function and this paper presents a good methodology to do it and a good system where dynamics of dopamine release can be tested at the level of behavior.

Weaknesses:

Key weaknesses have been resolved:

Receptor expression is consistent between time of the day and the authors picked two time points. The authors mention that the states of animals could affect LI (e.g. feeding state and anesthesia for sorting, see methods) were kept constant. These data and discussion are helpful.Giant fiber system is argued to be a great model and authors have added additional references. However I am not very deeply familiar with these references or the giant fiber system so I am not completely clear but the argument seems reasonable.The revised manuscript, shows data in the γ KCs (Figure 4C, Figure 5 - figure supplement 1) in addition to α/β KCs, so it appears there is consistency between lobes.The new data for Dop1R1 and Dop2R in MBON-γ1pedc>αβ helps with thinking about dopamine receptor co-localization and it would be a herculean talk to do this for all the regions but still keeps room open for different scenarios.

The papers discussion has been expanded to account for different possibilities which will help the readers get a more complete picture. I appreciate the review efforts and detailed response to reviewer comments.

---

## [Referee Report · Reviewer #2 (Public review)]

Summary:

Hiramatsu et al. investigated how cognate neurotransmitter receptors with antagonizing downstream effects localize within neurons when co-expressed. They focus on mapping the dopaminergic Dop1R1 and Dop2R receptors, corresponding to the mammalian D1- and D2-like dopamine receptors, which have opposing effects on intracellular cAMP levels, in neurons of the *Drosophila* mushroom body (MB). To visualize specific receptors in single neuron types within the crowded MB neuropil, the authors use existing dopamine receptor alleles tagged with 7 copies of split GFP to target the reconstitution of GFP tags specifically in the neurons of interest, providing a readout of receptor localization.

The authors demonstrate that both Dop1R1 and Dop2R are enriched, to differing degrees, in the axonal compartments of Kenyon cells cholinergic presynaptic inputs and in different dopamine neurons (DANs) that project axons to the MB. Co-localization studies of dopamine receptors with the presynaptic marker Brp suggest that Dop1R1, and to a greater extent Dop2R, localize near release sites. This pattern in DANs suggests Dop1R1 and Dop2R serve as dual-feedback autoreceptors. Finally, they provide evidence that the balance of Dop1R1 and Dop2R in the axons of two different DAN populations is differentially modulated by starvation, which plays a role in regulating appetitive behaviors.

In their revised manuscript, Hiramatsu et al. revisited the localization and functional integrity of Dop1R1 and Dop2R within the *Drosophila* mushroom body. This revision strengthens their claims with new high-resolution imaging data and additional behavioral assays, supporting the functional integrity of 7X split GFP-tagged receptors and their distinct localizations within neural circuits.

The revised manuscript by Hiramatsu et al. demonstrates substantial improvements in experimental design and data presentation, effectively addressing concerns raised during the initial review. The addition of advanced imaging techniques and behavioral data confirms the functionality of tagged receptors, while providing deeper insights into their spatial and functional dynamics within neural circuits modulating responses to environmental changes like starvation. This study makes an important contribution to neuroscience, enhancing our understanding of dopamine receptor distribution in circuits underlying learning and memory.

Strengths:

The authors use reconstitution of GFP fluorescence of split GFP tags integrated at the endogenous locus of dopamine receptors, providing a precise readout of receptor localization. This method preserves endogenous transcriptional and post-transcriptional regulation, a critical feature for protein localization studies.

The choice of the *Drosophila* mushroom body as a model system is excellent, as it is well-studied, its connectome is carefully reconstructed, and its role in behaviors and associative memory enables linking receptor localization patterns to circuit function and behavior. This approach allows the authors to demonstrate that antagonizing dopamine receptors can act as autoreceptors within the axonal compartments of MB-innervating DANs. Moreover, they show that starvation differentially modulates the balance of these receptors in distinct DANs, highlighting the role of this regulation in circuit function and behavior.

The incorporation of higher-resolution Airyscan microscopy and functional assays in the revision provide evidence that tagged receptors retain functionality and predominantly localize at presynaptic sites within Kenyon cells and DANs. These findings support the dual autoreceptor feedback model proposed.

Weaknesses:

While the revision significantly strengthens the manuscript, the absence of specific antibodies against these receptors remains a limitation. This is understandable given the challenges of generating antibodies against such proteins. However, the use of more direct validation methods, such as specific antibodies (if available), and employing higher-resolution techniques like expansion microscopy, could further validate and enhance the robustness of the findings.

---

## [Author Response]

The following is the authors’ response to the original reviews.

**eLife Assessment**
This study uses state-of-the-art methods to label endogenous dopamine receptors in a subset of *Drosophila* mushroom body neuronal types. The authors report that DopR1 and Dop2R receptors, which have opposing effects in intracellular cAMP, are present in axons termini of Kenyon cells, as well as those of two classes of dopaminergic neurons that innervate the mushroom body indicative of autocrine modulation by dopaminergic neurons. Additional experiments showing opposing effects of starvation on DopR1 and DopR2 levels in mushroom body neurons are consistent with a role for dopamine receptor levels increasing the efficiency of learned food-odour associations in starved flies. Supported by solid data, this is a valuable contribution to the field.

We thank the editors for the assessment, but request to change “DopR2” to “Dop2R”. The dopamine receptors in *Drosophila* have confusing names, but what we characterized in this study are called Dop1R1 (according to the Flybase; aka DopR1, dDA1, Dumb) and Dop2R (ibid; aka Dd2R). DopR2 is the name of a different dopamine receptor.

**Public Reviews:**

**Reviewer #1 (Public Review):**
Summary:This is an important and interesting study that uses the split-GFP approach. Localization of receptors and correlating them to function is important in understanding the circuit basis of behavior.Strengths:The split-GFP approach allows visualization of subcellular enrichment of dopamine receptors in the plasma membrane of GAL4-expressing neurons allowing for a high level of specificity.The authors resolve the presynaptic localization of DopR1 and Dop2R, in "giant" *Drosophila* neurons differentiated from cytokinesis-arrested neuroblasts in culture as it is not clear in the lobes and calyx.Starvation-induced opposite responses of dopamine receptor expression in the PPL1 and PAM DANs provide key insights into models of appetitive learning.Starvation-induced increase in D2R allows for increased negative feedback that the authors test in D2R knockout flies where appetitive memory is diminished.This dual autoreceptor system is an attractive model for how amplitude and kinetics of dopamine release can be fine-tuned and controlled depending on the cellular function and this paper presents a good methodology to do it and a good system where the dynamics of dopamine release can be tested at the level of behavior.Weaknesses:LI measurements of Kenyon cells and lobes indicate that Dop2R was approximately twice as enriched in the lobe as the average density across the whole neuron, while the lobe enrichment of Dop1R1 was about 1.5 times the average, are these levels consistent during different times of the day and the state of the animal. How were these conditions controlled and how sensitive are receptor expression to the time of day of dissection, staining, etc.

To answer this question, we repeated the experiment in two replicates at different times of day and confirmed that the receptor localization was consistent (Figure 3 – figure supplement 1); LI measurements showed that Dop2R is enriched more in the lobe and less in the calyx compared to Dop1R1 (Figure 3D). The states of animals that could affect LI (e.g. feeding state and anesthesia for sorting, see methods) were kept constant.

The authors assume without discussion as to why and how presynaptic enrichment of these receptors is similar in giant neurons and MB.

In the revision, we added a short summary to recapitulate that the giant neurons exhibit many characteristics of mature neurons (Lines #152-156): "Importantly, these giant neurons exhibit characteristics of mature neurons, including firing patterns (Wu et al., 1990; Yao & Wu, 2001; Zhao & Wu, 1997) and acetylcholine release (Yao et al., 2000), both of which are regulated by cAMP and CaMKII signaling (Yao et al., 2000; Yao & Wu, 2001; Zhao & Wu, 1997)." In addition, we found punctate Brp accumulations localized to the axon terminals of the giant neurons (former Figure 4D and 4E). Therefore, the giant neuron serves as an excellent model to study the presynaptic localization of dopamine receptors in isolated large cells.

Figures 1-3 show the expensive expression of receptors in alpha and beta lobes while Figure 5 focusses on PAM and localization in γ and β' projections of PAM leading to the conclusion that presynaptic dopamine neurons express these and have feedback regulation. Consistency between lobes or discussion of these differences is important to consider.

In the revised manuscript, we show data in the γ KCs (Figure 4C, Figure 5 - figure supplement 1) in addition to α/β KCs, and demonstrate the consistent synaptic localization of Dop1R1 and Dop2R as in α/β KCs (Figure 4B and 5A).

Receptor expression in any learning-related MBONs is not discussed, and it would be intriguing as how receptors are organized in those cells. Given that these PAMs input to both KCs and MBONs these will have to work in some coordination.

The subcellular localization of dopamine receptors in MBONs indeed provides important insights into the site of dopaminergic signaling in these neurons (Takemura et al., 2017; Pavlowsky et al., 2018; Pribbenow et al., 2022). Therefore, we added new data for Dop1R1 and Dop2R in MBON-γ1pedc>αβ (Figure 6). Interestingly, these receptors are localized to in the dendritic projection in the γ1 compartment as well as presynaptic boutons (Figure 6).

Although authors use the D2R enhancement post starvation to show that knocking down receptors eliminated appetitive memory, the knocking out is affecting multiple neurons within this circuit including PAMs and KCs. How does that account for the observed effect? Are those not important for appetitive learning?

In the appetitive memory experiment (Figure 9C), we knocked down Dop2R only in the select neurons of the PPL1 cluster, and this manipulation does not directly affect Dop2R expression in PAMs and KCs.

Starvation-induced enhancement of Dop2R expression in the PPL1 neurons (Figure 8F) would attenuate their outputs and therefore disinhibit expression of appetitive memory in starved flies (Krashes et al., 2009). Consistently, Dop2R knock-down in PPL1 impaired appetitive memory in starved flies (Figure 9C). We revised the corresponding text to make this point clearer (Lines #224227).

The evidence for fine-tuning is completely based on receptor expression and one behavioral outcome which could result from many possibilities. It is not clear if this fine-tuning and presynaptic feedback regulation-based dopamine release is a clear possibility. Alternate hypotheses and outcomes could be considered in the model as it is not completely substantiated by data at least as presented.

The reviewer’s concern is valid, and the presynaptic dopamine tuning by autoreceptors may need more experimental support. We therefore additionally discussed another possibility (Lines #289-291): “Alternatively, these presynaptic receptors could potentially receive extrasynaptic dopamine released from other DANs. Therefore, the autoreceptor functions need to be experimentally clarified by manipulating the receptor expression in DANs.”

**Reviewer #2 (Public Review):**
Summary:Hiramatsu et al. investigated how cognate neurotransmitter receptors with antagonizing downstream effects localize within neurons when co-expressed. They focus on mapping the localization of the dopaminergic Dop1R1 and Dop2R receptors, which correspond to the mammalian D1- and D2-like dopamine receptors, which have opposing effects on intracellular cAMP levels, in neurons of the *Drosophila* mushroom body (MB). To visualize specific receptors in single neuron types within the crowded MB neuropil, the authors use existing dopamine receptor alleles tagged with 7 copies of split GFP to target reconstitution of GFP tags only in the neurons of interest as a read-out of receptor localization. The authors show that both Dop1R1 and Dop2R, with differing degrees, are enriched in axonal compartments of both the Kenyon Cells cholinergic presynaptic inputs and in different dopamine neurons (DANs), which project axons to the MB. Co-localization studies of dopamine receptors with the presynaptic marker Brp suggest that Dop1R1 and, to a larger extent Dop2R, localize in the proximity of release sites. This localization pattern in DANs suggests that Dop1R1 and Dop2R work in dual-feedback regulation as autoreceptors. Finally, they provide evidence that the balance of Dop1R1 and Dop2R in the axons of two different DAN populations is differentially modulated by starvation and that this regulation plays a role in regulating appetitive behaviors.Strengths:The authors use reconstitution of GFP fluorescence of split GFP tags knocked into the endogenous locus at the C-terminus of the dopamine receptors as a readout of dopamine receptor localization. This elegant approach preserves the endogenous transcriptional and post-transcriptional regulation of the receptor, which is essential for studies of protein localization.The study focuses on mapping the localization of dopamine receptors in neurons of the mushroom body. This is an excellent choice of system to address the question posed in this study, as the neurons are well-studied, and their connections are carefully reconstructed in the mushroom body connectome. Furthermore, the role of this circuit in different behaviors and associative memory permits the linking of patterns of receptor localization to circuit function and resulting behavior. Because of these features, the authors can provide evidence that two antagonizing dopamine receptors can act as autoreceptors within the axonal compartment of MB innervating DANs. The differential regulation of the balance of the two receptors under starvation in two distinct DAN innervations provides evidence of the role that regulation of this balance can play in circuit function and behavioral output.Weaknesses:The approach of using endogenously tagged alleles to study localization is a strength of this study, but the authors do not provide sufficient evidence that the insertion of 7 copies of split GFP to the C terminus of the dopamine receptors does not interfere with the endogenous localization pattern or function. Both sets of tagged alleles (1X Venus and 7X split GFP tagged) were previously reported (Kondo et al., 2020), but only the 1X Venus tagged alleles were further functionally validated in assays of olfactory appetitive memory. Despite the smaller size of the 7X split-GFP array tag knocked into the same location as the 1X venus tag, the reconstitution of 7 copies of GFP at the C terminus of the dopamine receptor, might substantially increase the molecular bulk at this site, potentially impeding the function of the receptor more significantly than the smaller, single Venus tag. The data presented by Kondo et al. 2020, is insufficient to conclude that the two alleles are equivalent.

In the revision, we validated the function of these engineered receptors by a new set of olfactory learning experiments. Both these receptors in KCs were shown to be required for aversive memory (Kim et al., 2007, Scholz-Kornehl et al., 2016). As in the anatomical experiments, we induced GFP110 expression in KC of the flies homozygous for 7xGFP_11_-tagged receptors using *MB-Switch* and 3 days of RU486 feeding o. We confirmed STM performance of these flies were not significantly different from the control (Figure 2 – figure supplement 1). Thus, these fusion receptors are functional.

The authors' conclusion that the receptors localize to presynaptic sites is weak. The analysis of the colocalization of the active zone marker Brp whole-brain staining with dopamine receptors labeled in specific neurons is insufficient to conclude that the receptors are localized at presynaptic sites. Given the highly crowded neuropil environment, the data cannot differentiate between the receptor localization postsynaptic to a dopamine release site or at a presynaptic site within the same neuron. The known distribution of presynaptic sites within the neurons analyzed in the study provides evidence that the receptors are enriched in axonal compartments, but co-labeling of presynaptic sites and receptors in the same neuron or super-resolution methods are needed to provide evidence of receptor localization at active zones. The data presented in Figures 5K-5L provides compelling evidence that the receptors localize to neuronal varicosities in DANs where the receptors could play a role as autoreceptors.Given the highly crowded environment of the mushroom body neuropil, the analysis of dopamine receptor localization in Kenyon cells is not conclusive. The data is sufficient to conclude that the receptors are preferentially localizing to the axonal compartment of Kenyon cells, but co-localization with brain-wide Brp active zone immunostaining is not sufficient to determine if the receptor localizes juxtaposed to dopaminergic release sites, in proximity of release sites in Kenyon cells, or both.

To better resolve the microcircuits of KCs, we triple-labeled the plasma membrane and DAR::rGFP in KCs, and Brp, and examined their localizations with high-resolution imaging with Airyscan. This strategy revealed the receptor clusters associated with Brp accumulation within KCs (Figure 4). To further verify the association of DARs and active zones within KCs, we co-expressed Brp^short^::mStraw and GFP_1-10_ and confirmed their colocalization (Figure 5A), suggesting presynaptic localization of DARs in KCs. With these additional characterizations, we now discuss the significance of receptors at the presynaptic sites of KCs.

**Recommendations for the authors:**

**Reviewer #1 (Recommendations For The Authors):**
This is an important and interesting study that uses the split-GFP approach. Localization of receptors and correlating them to function is important in understanding the circuit basis of behavior.For Figure 1, the authors show PAM, PPL1 neurons, and the ellipsoid body as a validation of their tools (Dop1R1-T2A-GAL4 and Dop2R-T2A-GAL4) and the idea that these receptors are colocalized. However, it appears that the technique was applied to the whole brain so it would be great to see the whole brain to understand how much labelling is specific and how stochastic. Methods could include how dissection conditions were controlled and how sensitive are receptor expression to the time of day of dissection, staining, etc.

The expression patterns of the receptor T2A-GAL4 lines (Figure 1A and 1B) are consistent in the multiple whole brains (Kondo et al., 2020, Author response image 1).

The significance of the expression of these two receptors in an active zone is not clearly discussed and presynaptic localization is not elaborated on. Would something like expansion microscopy be useful in resolving this? It would be important to discuss that as giant neurons in culture don't replicate many aspects of the MB system.

In the revised manuscript, we elaborated discussion regarding the function of the two antagonizing receptors at the AZ (Lines #226-275).

Does MB-GeneSwitch > GFP1-1 reliably express in gamma lobes? Most of the figures show alpha/beta lobes.

Yes. MB-GeneSwitch is also expressed in γ KCs, but weakly. 12 hours of RU486 feeding, which we did in the previous experiments, was insufficient to induce GFP reconstitution in the γ KCs. By extending the time of transgene induction, we visualized expression of Dop1R1 and Dop2R more clearly in γ KCs. Their localization is similar to that in the α/β KCs (Figure 4C, Figure 5 - figure supplement 1).

Figure 6, y-axis says protein level. At first, I thought it was related to starvation so maybe authors can be more specific as the protein level doesn't indicate any aspect of starvation.

We appreciate this comment, and the labels on the y-axis were now changed to “rGFP levels” (Figure 8C and 8F, Figure 8 - figure supplement 1B, 1D and 1F).

**Reviewer #2 (Recommendations For The Authors):**
Title:The title of the manuscript focuses on the tagging of the receptors and their synaptic enrichment.Given that the alleles used in the study were generated in a previously published study (Kondo et al, 2020), which describes the receptor tagging and that the data currently provided is insufficient to conclude that the receptors are localizing to synapses, the title should be changed to reflect the focus on localizing antagonistic cognate neurotransmitter receptors in the same neuron and their putative role as autoreceptors in DANs.

Following this advice, we removed the methodology from the title and revised it to “Synaptic enrichment and dynamic regulation of the two opposing dopamine receptors within the same neurons”.

Minor issues with text and figures:Figure 1A conclusion from Figure 1 is that the two receptors are co-expressed in Kenyon cells. Please provide panels equivalent to the ones shown in D-G, with Kenyon cells cell bodies, or mark these cells in the existing panels, if present. Line 111 refers to panel 1D as the Kenyon cells panel, which is currently a PAM panel.

We added images for coexpression of these receptors in the cell bodies of KCs (Figure 1 - figure supplement 1) and revised the text accordingly (Lines #89-90).

Given that most of the study centers on visualizing receptor localization, it would benefit the reader to include labels in Figure 1 that help understand that these panels reflect expression patterns rather than receptor localization. For instance, rCD2::GFP could be indicated in the Dop1R1-LexA panels.

As suggested, labels were added to indicate the UAS and lexAop markers (Figure 1D, 1E, 1G-1I and Figure 1 – figure supplement 1).

Given that panels D-E focus on the cell bodies of the neurons, it could be beneficial for the reader to present the ellipsoid body neurons using a similar view that only shows the cell bodies. Similarly, one could just show the glial cell bodies .

We now show the cell bodies of ring neurons (Figure 1G) and ensheathing glia (Figure 1I).

For panel 1E, please indicate the subset of PPL1 neurons that both expressed Dop1R1 and Dop2R, as indicated in the text, as it is currently unclear from the image.

*Dop1R1-T2A-LexA* was barely detected in all PPL1 (Figure 1E). We corrected the confusing text (Lines #95-96).

Figure 2The cartoon of the cell-type-specific labeling should show that the tag is 7XFP-11 and the UAScomponent FP-10, as the current cartoon leads the reader to conclude that the receptors are tagged with a single copy of split GFP. The detail that the receptors are tagged with 7 copies of split GFP is only provided through the genotype of the allele in the resource table. This design aspect should be made clear in the figure and the text when describing the allele and approach used to tag receptors in specific neuron types.

We now added the construct design in the scheme (Figure 2A) and revised the corresponding text (Line #101-103).

Panel A. The arrow representing the endogenous promoter in the yellow gene representation should be placed at the beginning of the coding sequence. Currently, the different colors of what I assume are coding (yellow) and non-coding (white) transcript regions are not described in the legend. I would omit these or represent them in the same color as thinner boxes if the authors want to emphasize that the tag is inserted at the C terminus within the endogenous locus.

The color scheme was revised to be more consistent and intuitive (Figure 2A).

Figure 3Labels of the calyx and MB lobes would benefit readers not as familiar with the system used in the study. In addition, it would be beneficial to the reader to indicate in panel A the location of the compartments analyzed in panel H (e.g., peduncle, α3).

Figure 3A was amended to clearly indicate the analyzed MB compartments.

Adding frontal and sagittal to panels B-E, as in Figure 2, would help the reader interpret the data.

In Figure 3B, “Frontal” and “Sagittal” were indicated.

Panel F-G. A scale bar should be provided for the data shown in the insets. Could the author comment on the localization of Dop1R1 in KCs? The data in the current panel suggests that only a subset of KCs express high levels of receptors in their axons, as a portion of the membrane is devoid of receptor signals. This would be in line with differential dopamine receptor expression in subsets of Kenyon cells, as shown in Kondo et al., 2020, which is currently not commented on in the paper.

We confirmed that the majority of the KCs express both *Dop1R1* and *Dop2R* genes (Figure 1 - figure supplement 1). LIs should be compared within the same cells rather than the differences of protein levels between cell types as they also reflect the GAL4 expression levels.

Panel H. Some P values are shown as n.s. (p> 0.05). Other non-significant p values in this panel and in other figures throughout the paper are instead reported (e.g. peduncle P=0.164). For consistency, please report the values as n.s. as indicated in the methods for all non-significant tests in this panel and throughout the manuscript.

We now present the new dataset, and the graph represents the appropriate statistical results (Figure 3D; see the methods section for details).

The methods of labeling the receptors through the expression of the GeneSwitch-controlled GFP1-10 in Kenyon cells induced by RU486 are not provided in the methods. Please provide a description of this as referenced in the figure legend and the genotypes used in the analysis shown in the panels.

The method of RU486 feeding has been added. We apologize for the missing method.

Figure 4Please provide scale bars for the inset in panels A-B.

Scale bars were added to all confocal images.

The current analysis cannot distinguish between postsynaptic and presynaptic dopamine receptors in KCs, and the figure title should reflect this.

We now present the new data dopamine receptors in KCs and clearly distinguish Brp clusters of the KCs and other cell types (Figure 4, Figure 5).

The reader could benefit from additional details of using the giant neuron model, as it is not commonly used, and it is not clear how to relate this to interpret the localization of dopaminergic receptors within Kenyon cells. The use of the venus-tagged receptor variant should be introduced in the text, as using a different allele currently lacks context. Figures 4F-4J show that the receptor is localizing throughout the neuron. Quantifying the fraction of receptor signal colocalizing with Brp could aid in interpreting the data. However, it would still not be clear how to interpret this data in the context of understanding the localization of the receptors in neurons within fly brain circuits. In the absence of additional data, the data provided in Figure 4 is inconclusive and could be omitted, keeping the focus of the study on the analysis of the two receptors in DANs. Co-expressing a presynaptic marker in Kenyon cells (e.g., by expressing Brp::SNAP) in conjunction with rGFP labeled receptor would provide additional evidence of the relationship of release sites in Kenyon cells and tagged dopamine receptors in these same cells and could add evidence in support to the current conclusion.

Following the advice, we added a short summary to recapitulate that the giant neurons exhibit many characteristics of mature neurons (Lines #152-156): "Importantly, these giant neurons exhibit characteristics of mature neurons, including firing patterns (Wu et al., 1990; Yao & Wu, 2001; Zhao & Wu, 1997) and acetylcholine release (Yao et al., 2000), both of which are regulated by cAMP and CaMKII signaling (Yao et al., 2000; Yao & Wu, 2001; Zhao & Wu, 1997)." Therefore, the giant neuron serves as an excellent model to study the presynaptic localization in large cells in isolation.

To clarify polarized localization of Brp clusters and dopamine receptors but not "localizing throughout the neuron", we now show less magnified data (Figure 5C). It clearly demonstrates punctate Brp accumulations localized to the axon terminals of the giant neurons (former Figure 4D and 4E). This is the same membrane segment where Dop1R1 and Dop2R are localized (Figure 5C). Therefore, the association of Brp clusters and the dopamine receptors in the isolated giant neurons suggests that the subcellular localization in the brain neurons is independent of the circuit context.

As the giant neurons do not form intermingled circuits, venus-tagged receptors are sufficient for this experiment and simpler in genetics.

Following the suggestion to clarify the AZ association of the receptors in KCs, we coexpressed Brpshort-mStraw and GFP1-10 in KCs and confirmed their colocalization (Figure 5A).

Figure 6The data and analysis show that starvation induces changes in the α3 compartment in PPL1 neurons only, while the data provided shows no significant change for PPL1 neurons innervating other MB compartments. This should be clearly stated in lines 174-175, as it is implied that there is a difference in the analysis for compartments other than α3. Panel L of Figure 6 - supplement 1 shows no significant change for all three compartments analyzed and should be indicated as n.s. in all instances, as stated in the methods.

We revised the text to clarify that the starvation-induced differences of Dop2R expression were not significant (Lines #217-219). The reason to highlight the α3 compartment is that both Dop1R1 and Dop2R are coexpressed in this PPL1 neuron (Figure 8D).

Additional minor comments:There are a few typos and errors throughout the manuscript. The text should be carefully proofread to correct these. Here are the ones that came to my attention:Please reference all figure panels in the text. For instance, Figure 3A is not mentioned and should be revised in line 112 as Figure 3A-E.Lines 103-104. The sentence "LI was visualized as the color of the membrane signals" is unclear and should be revised.Figure 4 legend - dendritic claws should likely be B and C and not B and E.Lines 147 - Incorrect figure panels, should be 5C-L or 5D-E.Line 241 - DNAs should be DANs.Methods - please define what the abbreviation CS stands for.

We really appreciate for careful reading of this reviewer. All these were corrected.